# Accurate detection of identity-by-descent segments in human ancient DNA

Harald Ringbauer [1,2,9] ✉, Yilei Huang[1,3,9], Ali Akbari[2,4,5], Swapan Mallick[4,5,6], Iñigo Olalde [2,7,8], Nick Patterson[2,5] & David Reich [2,4,5,6] ✉

Long DNA segments shared between two individuals, known as identity-by-descent (IBD), reveal recent genealogical connections. Here we introduce ancIBD, a method for identifying IBD segments in ancient human DNA (aDNA) using a hidden Markov model and imputed genotype probabilities. We demonstrate that ancIBD accurately identifies IBD segments >8 cM for aDNA data with an average depth of >0.25× for whole-genome sequencing or >1× for 1240k single nucleotide polymorphism capture data. Applying ancIBD to 4,248 ancient Eurasian individuals, we identify relatives up to the sixth degree and genealogical connections between archaeological groups. Notably, we reveal long IBD sharing between Corded Ware and Yamnaya groups, indicating that the Yamnaya herders of the Pontic-Caspian Steppe and the Steppe-related ancestry in various European Corded Ware groups share substantial co-ancestry within only a few hundred years. These results show that detecting IBD segments can generate powerful insights into the growing aDNA record, both on a small scale relevant to life stories and on a large scale relevant to major cultural-historical events.

Some pairs of individuals share long, nearly identical genomic segments, so-called IBD segments, that must be co-inherited from a recent common ancestor because recombination during each meiosis leads to the rapid break-up of these segments. Consequently, long IBD segments provide an ideal signal to probe recent genealogical connections and have been used as a distinctive signal for a range of downstream applications such as identifying biological relatives or inferring recent demography[1–3]. Several existing methods identify IBD segments for single nucleotide polymorphism (SNP) array or whole-genome sequence data[4–6] but they require confident diploid genotype calls. These are not achievable for most human aDNA data because of too low genomic coverage (<5× average coverage per site) and comparably high error rates due to degraded and short DNA molecules. So far only a few exceptional applications of IBD

to comparably high-quality aDNA have been published[7,8]. First efforts to identify IBD on the basis of imputed data have been fruitful[9–12] but those require higher coverage not routinely available for aDNA. Importantly, they do not include a systematic evaluation of the IBD calling pipelines, a critical task given that IBD detection accuracy is expected to decay for short segments and low-coverage data. Practical downstream applications, such as demographic modelling, require information about power, length biases and false positive rates either to account directly for these error processes or to identify thresholds of data quality.

   Here, we present and systematically evaluate ancIBD, a method to detect IBD segments in human aDNA data. In brief, ancIBD starts from phased genotype likelihoods imputed by GLIMPSE[13], which are then screened using a hidden Markov model (HMM) to infer IBD blocks

[1]Department of Archaeogenetics, Max Planck Institute for Evolutionary Anthropology, Leipzig, Germany. [2]Department of Human Evolutionary Biology, Harvard University, Cambridge, MA, USA. [3]Bioinformatics Group, Institute of Computer Science, Universität Leipzig, Leipzig, Germany. [4]Department of Genetics, Harvard Medical School, Boston, MA, USA. [5]Broad Institute of Harvard and MIT, Cambridge, MA, USA. [6]Howard Hughes Medical Institute, Harvard Medical School, Boston, MA, USA. [7]BIOMICs Research Group, University of the Basque Country, Vitoria-Gasteiz, Spain. [8]Ikerbasque-Basque Foundation of Science, Bilbao, Spain. [9]These authors contributed equally: Harald Ringbauer, Yilei Huang. ✉e-mail: harald_ringbauer@eva.mpg.de; reich@genetics.med.harvard.edu

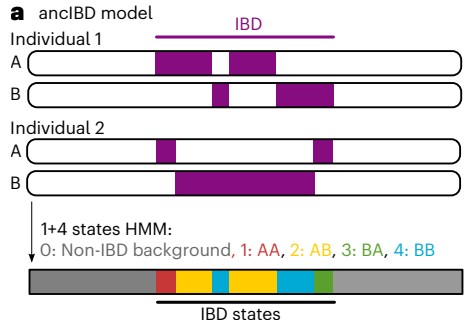

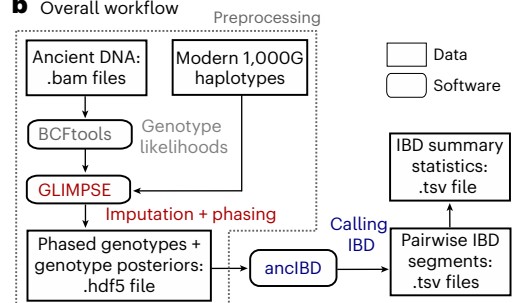

**Fig. 1 | Overview of the ancIBD algorithm. a**, Sketch of the ancIBD HHM. The HMM has five states: one background state of no allele sharing and four states modelling the four possible IBD-sharing states between two phased diploid genomes. We model phase switch errors within a true IBD segment as a transition between the four IBD states. **b**, Visualization of the full pipeline to call IBD. First, aDNA data are imputed and phased using GLIMPSE and a panel of modern reference haplotypes. We note that users can customize these upstream steps; for example, use other tools to obtain genotype likelihoods or use different reference panels. Our core software (ancIBD) is then applied to the imputed data to screen for IBD. It produces two tables, one listing all inferred IBD segments and one listing IBD summary statistics for each pair of individuals.

(Fig. 1). We then identified default parameters that optimize performance on so-called 1240k capture data. This set of ~1.1 million autosomal SNPs is targeted by in-solution enrichment experiments that have produced more than 70% of genome-wide human aDNA datasets to date[14–16]. Our tests show that ancIBD robustly identifies IBD longer than 8 cM in aDNA data—for SNP capture with at least 1x average coverage depth (calculated on SNP target) and for whole-genome sequencing (WGS) as low as 0.25× average genomic coverage.

## Results

### Identifying IBD with ancIBD

Our method consists of two computational steps (Fig. 1b). In a pre-processing step, the aDNA data are first computationally imputed and phased using a modern reference haplotype panel. In the main step, we apply a custom HMM to identify IBD segments.

For the preprocessing, we use imputation software that has been shown to work well for low-coverage data, GLIMPSE[13], which we apply to aligned sequence data (in .bam format) to impute genotype likelihoods at the 1240k sites, using haplotypes in the 1000 Genome Project as the reference panel[17]. Our full imputation pipeline is described in Supplementary Note 3. Previous evaluation of imputing aDNA data this way showed that imputed common variants, which are highly informative about IBD sharing, are of good quality down to mean coverage depth as low as 0.5–1.0× (refs. 18,19).

The details of the main ancIBD HMM are described in Methods. Briefly, the HMM is based on a total of five hidden states, where one state models non-IBD and four states the possible ways of IBD sharing between two phased genomes (Fig. 1a). The emission probabilities are based on the imputed posterior genotype probability and phasing. The standard forward–backward algorithm[20] yields the posterior probability of being in one of the four IBD states, which is postprocessed to obtain the final IBD segment calls.

### Evaluating ancIBD

We performed two sets of experiments to evaluate the quality of IBD calls of ancIBD at various sequencing depths. First, we copied IBD segments of known length into pairs of genomes (Methods). Second, we downsampled high-coverage empirical aDNA data.

**Performance on copied-in IBD segments.** When applying ancIBD to the simulated data with copied-in IBD (simulation procedures are described in Supplementary Note 2 and visualized in Extended Data Fig. 1), we observed that the inferred IBD segments remain accurate and that their length distribution peaks around the true value for WGS data down to about 0.25× coverage and for 1240k capture data down

to 1× coverage at 1240k sites (Fig. 2). We found that ancIBD on average overestimates the length of IBD segments but in the recommended coverage cutoff the length errors remain within ~1 cM (Extended Data Tables 1 and 2).

**Performance on downsampled aDNA data.** To assess performance on downsampled empirical aDNA data, we used four high-coverage genomes of ancient individuals, all ~5,000 years old and associated with the Southern Siberian Afanasievo culture (Supplementary Note 5)[21]. When comparing the IBD calls in the downsampled data to the IBD calls of the original high-coverage data, we found that WGS substantially outperforms 1240k data of the same coverage. For long IBD segments (>10 cM) that are particularly informative when detecting relatives, ancIBD achieves high precision and recall (>90%) for all coverages tested here (WGS data 0.1× to 5×; 1240k data 0.5× to 2×). For intermediate range segments (8–10 cM), ancIBD maintains reasonable recall (~80%) at all coverages while having less than 80% precision at 0.5× for 1240k data. Overall, ancIBD yields accurate IBD calling (~90% or higher precision) at >0.25× WGS data and >1× 1240k data (Extended Data Fig. 2).

**Comparing to other methods.** Several recent publications have applied softwares designed to detect IBD in high-quality present-day data on imputed aDNA data (for example, using GLIMPSE)[9,10]. We compared the performance of ancIBD to such methods, using the downsampled empirical aDNA data described above.

Softwares to call IBD can be classified into two categories, ones that require prior phasing and ones that use unphased data as input. The former search for long, identical haplotypes, while the latter primarily use, directly or implicitly, the signal of 'opposing homozygotes' (two samples being homozygous for different alleles), which are lacking in IBD segments.

In preliminary tests, we found that methods that require phasing information have very low power to detect IBD in imputed aDNA data, potentially because of high switch error rates in imputed ancient genomes[19], which is an order of magnitude higher than what is attainable for phasing Biobank-scale modern data[22].

Therefore, we focus our detailed comparison on two methods that do not require phasing information, IBIS[23] and IBDseq[24]. IBIS detects IBD segments by screening for genomic regions with few opposing homozygotes. Our results on downsampled aDNA data show that this method mostly maintains higher precision at the expense of a lower recall, particularly at lower coverages. Despite keeping precision at >90%, for segments >8 cM, IBIS recall drops to ~50% for ~1× 1240k data (Extended Data Fig. 2).

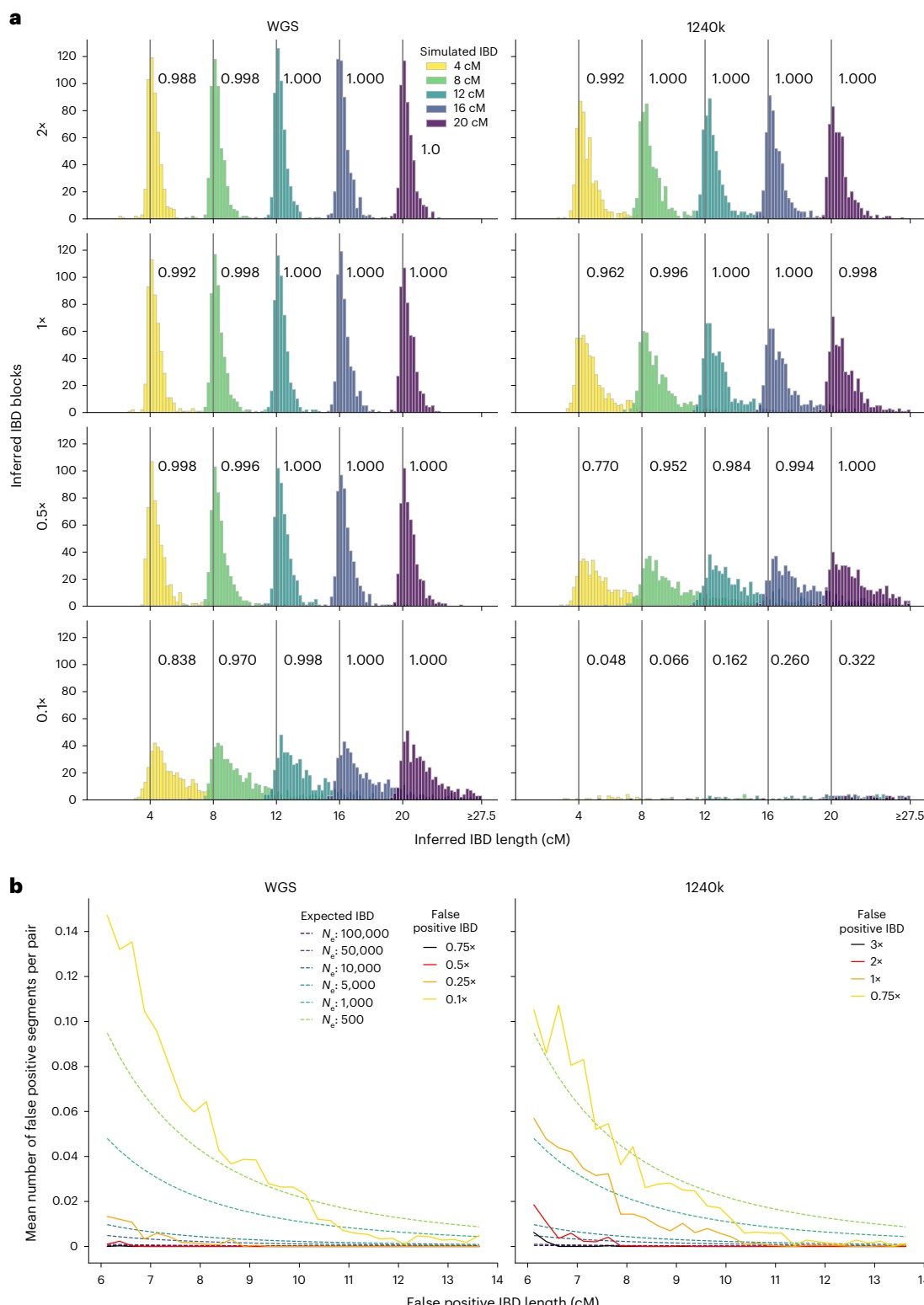

**Fig. 2 | Performance of ancIBD on simulated IBD segments. a**, Power and segment length errors. We copied-in IBD segments of lengths 4, 8, 12, 16 and 20 cM into synthetic diploid samples. We simulated shotgun-like and 1240k-like data (Supplementary Note 2) and visualize false positive, power and length bias for 2×, 1×, 0.5× and 0.25× coverage (rows). For each parameter set and IBD length, we simulated 500 replicates of pairs of chromosome 3, each pair with a single, randomly placed, copied-in IBD segment. The power (or recall) of detecting IBD segments of each simulated length is indicated in the text next to the corresponding grey vertical bar. Results for other coverages are shown in Supplementary Fig. 4. **b**, False positive rate. We downsampled high-quality empirical aDNA data without IBD segments (Supplementary Table 6) to establish false positive rates of IBD segments for various coverage and IBD lengths (Supplementary Note 7). The y axis shows the mean number of false positive IBD segments per pair of chromosome 3 in each length bin (bin width 0.25 cM). To contextualize these false positive rates, we also depict expected IBD sharing assuming various constant population sizes (dotted lines, calculated as described in ref. 58). If the false positive rate is on a similar order of magnitude or larger than expected for a population of that effective population size ($N_e$), individual IBD calls of that length for that coverage and demographic scenario are likely to be false positives.

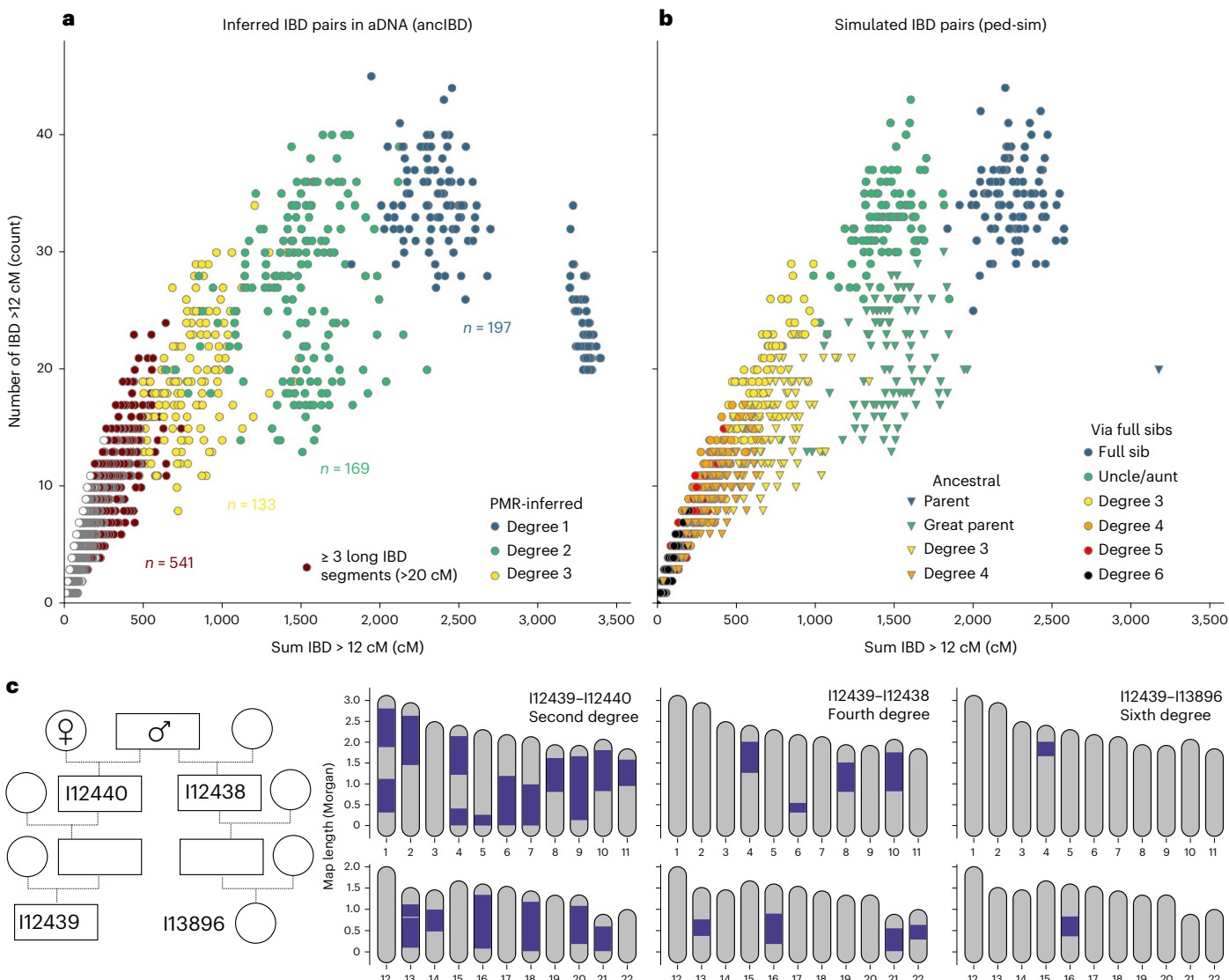

**Fig. 3 | Inferring biological relatives in the aDNA record using long IBD inferred with ancIBD. a**, Inferred IBD among pairs of 4,248 ancient Eurasian individuals. The plot visualizes both the count (*y* axis) as well as the summed length (*x* axis) of all IBD >12 cM long. For comparison, we colour-code pairs on the basis of relatedness estimates from pairwise mismatch rates (PMR) that can detect up to third-degree relatives (Supplementary Note 9). We also annotate new relatives found by ancIBD, indicated by at least three very long IBD segments (>20 cM) typical of up to sixth-degree relatives. **b**, Simulated IBD among pairs of relatives. For each relative class, we simulated 100 replicates using the software

ped-sim[25], as described in Supplementary Note 8. As in **a**, we depict the summed length and the count of all IBD at least 12 cM long. **c**, Inferred IBD among four ancient English Neolithic individuals, who lived about 5,700 years ago and were entombed at Hazleton North long cairn. A full pedigree was previously reconstructed using first- and second-degree relatives inferred using pairwise SNP matching rates[26]. We depict all IBD at least 12 cM long. The four individuals were genotyped using 1240k aDNA capture (I12438, 3.7× average coverage on target; I12440, 2.1×; I13896, 1.1×; I12439, 6.7×).

IBDseq was designed for WGS data. It works by computing likelihood ratios of IBD and non-IBD states for each marker and then identifies IBD segments by searching for regions with high cumulative scores. Our results on downsampled empirical ancient aDNA data indicate that precision and recall of IBDseq drop substantially at lower coverages, achieving <50% precision for ~1× 1240k data, a coverage regime typical for most aDNA samples (Supplementary Figs. 16 and 17).

**Detecting close and distant relatives with ancIBD**

To showcase the utility of IBD segments to detect biological relatives, we applied ancIBD to a set of 4,248 published ancient Eurasian individuals. Sample quality filtering and downstream bioinformatic processing are described in Methods. When plotting the total sum and the total count of IBD segments longer than 12 cM, we find that the pattern of

IBD sharing (Fig. 3a) closely mirrors simulated IBD sharing between various degrees of relatives (using the software ped-sim[25]) (Fig. 3b). A first-degree relative cluster becomes apparent, with a parent–offspring cluster (where the whole genome is in IBD) and a full-sibling cluster. The parent–offspring cluster in the simulated IBD dataset consists of one point, as expected because parent and offspring share each of the 22 chromosomes fully IBD. In the inferred IBD dataset, the apparent parent–offspring cluster is spread out more widely, including also individuals with more than 22 IBD segments—the reason for this is that sporadically very long IBD are broken up by artificial gaps and if they are too big they are not merged by the default gap merging of ancIBD. Overall this effect remains modest and in the parent–offspring cluster the total number of inferred IBD segments is in most cases only slightly elevated beyond the expected 22.

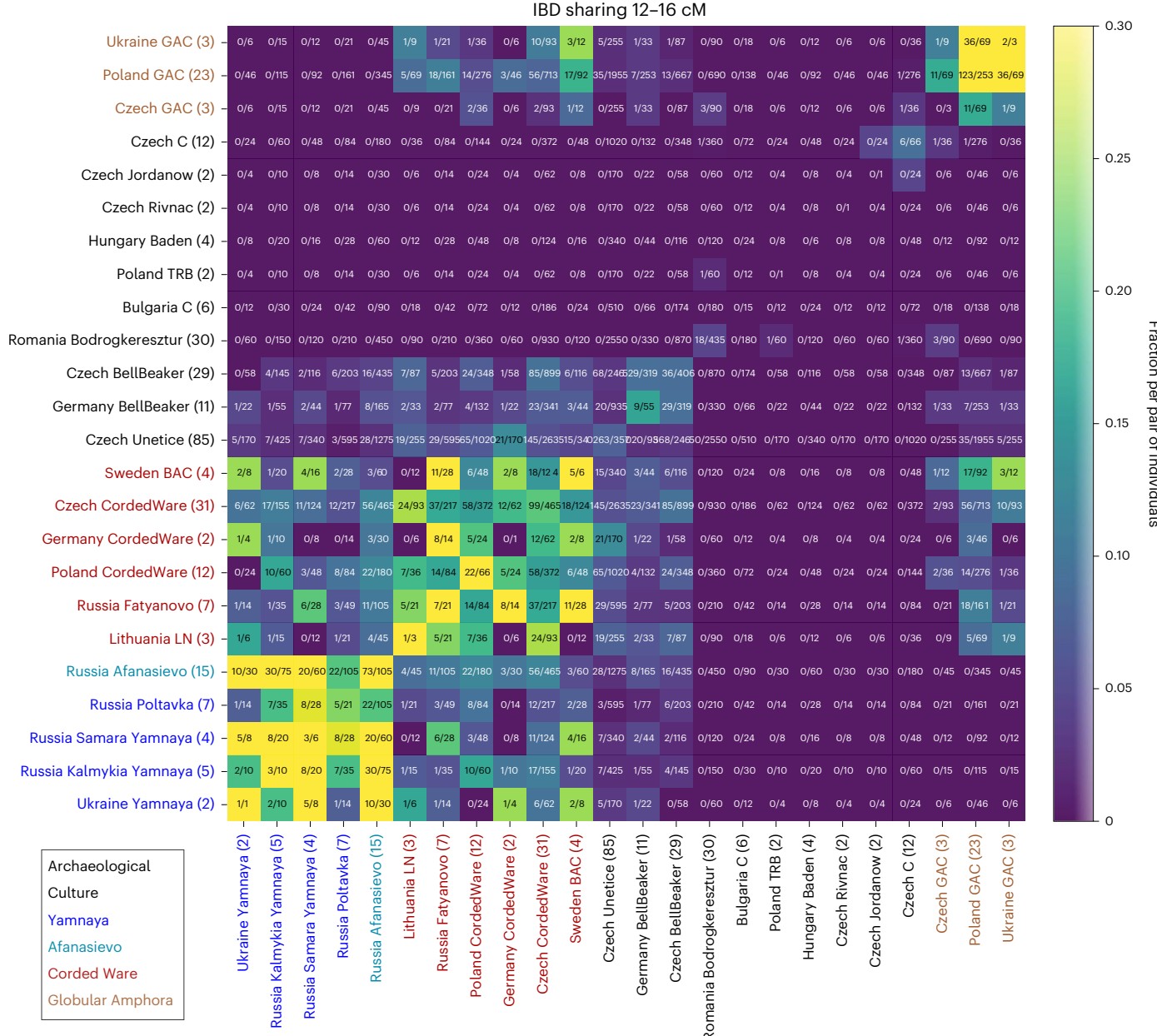

**Fig. 4 | Inferred IBD segments between various Eneolithic and Bronze Age West Eurasian Groups.** We visualize IBD segments 12–16 cM long (for IBD sharing in other length classes see Extended Data Fig. 3). We applied ancIBD to identify IBD segments between all pairs of 304 West Eurasian ancient individuals (all previously published data; Supplementary Table 3) organized into 24 archaeological groups. The number in the parenthesis indicates the sample size for each archaeological group. For each pair of groups, we plot the fraction of all possible pairs of individuals that share at least one IBD 12–16 cM long, which we obtained by dividing the total number of pairs that share such IBD segments by the total number of all possible pairs: between two different groups of $n_1$ and $n_2$ individuals, one has $n_1 n_2$ pairs, while within a group (on the diagonal in the figure) of size $n$ one has $n(n-1)/2$ pairs. LN, Late Neolithic; BAC, Battle Axe Culture; C, Chalcolithic; TRB, Trichterbecherkultur (Funnelbeaker culture); GAC, Globular Amphora Culture.

Further, we observe two clear second-degree relative clusters that correspond to biological great-parent grandchildren and aunt/uncle–niece/nephew relationships. Half-siblings are expected to form a gradient between these two clusters, with their average position depending on whether the shared parent is maternal (on average more but shorter shared segments) or paternal (fewer but longer shared segments)[25].

In the simulated data, IBD clusters for third-degree and more distant relatives increasingly overlap (Fig. 3b) and the empirical IBD distribution follows this gradient (Fig. 3a). Owing to this biological variation in genetic relatedness, it is not possible to uniquely assign individuals to specific relative clusters beyond third-degree relatives even if the exact IBD is known. However, these pairs with multiple long shared segments still unambiguously indicate very recent biological relatedness. Most biological relatives up to the sixth degree will share two or more long IBD segments[25]. For instance, we identified two long IBD segments in a sixth-degree relative from Neolithic Britain (Fig. 3c), a relationship that was previously reconstructed from a pedigree of first-degree and second-degree relatives identified using average pairwise genotype mismatch rates[26]. In most human populations, pairs of biologically unrelated (that is, related at most by tenth degree) individuals share only sporadically single IBD segments[27–29]. Thus, the sharing of many long IBD segments provides a distinct signal for identifying close genealogical relationships that we can detect with ancIBD.

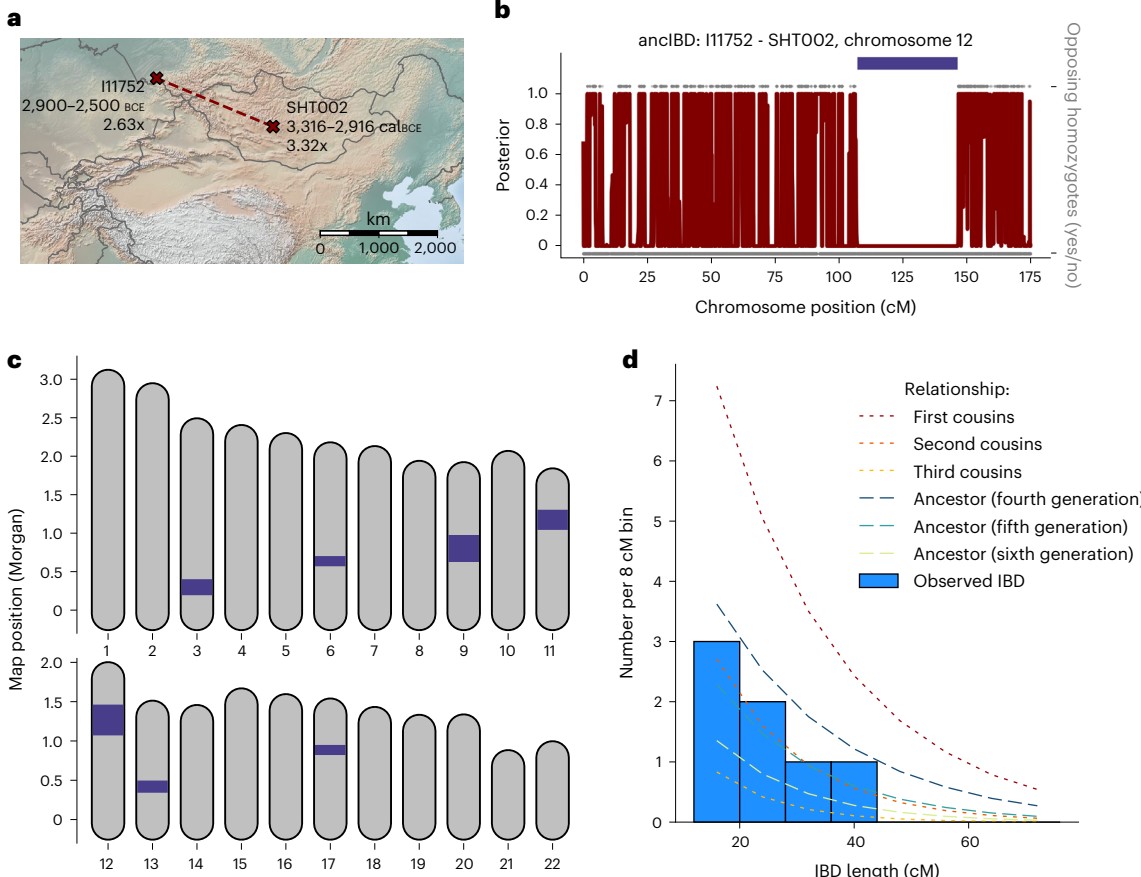

**Fig. 5 | A geographically distant pair of ancient biological relatives detected with ancIBD. a**, When screening ancient Eurasian individuals for IBD segments (Fig. 3), we detected a pair of biological relatives whose remains were buried 1,410 km apart, one in central Mongolia and one in Southern Russia. The two individuals were previously published in two different publications[35,59]. Both individuals are archaeologically associated with the Afanasievo culture and genetically cluster with other Afanasievo individuals[35,59]. **b**, Posterior of non-IBD state on chromosome 12, which has the longest inferred IBD segment (39.1 cM

long, indicated as a dark blue bar). We also plot opposing homozygotes (upper grey dots), whose absence is a necessary signal of IBD. Only SNPs where both markers have an imputed genotype probability >0.99 are plotted. **c**, Plot of all inferred IBD segments longer than 12 cM. **d**, Histogram of inferred IBD segment lengths, as well as theoretical expectations for various types of relatives (calculated using formulas described in ref. 29). Panels **b**–**d** were all created using default plotting functions bundled into the ancIBD software package.

## Recent links among Eneolithic and Bronze Age groups

Because recombination acts as a rapid clock (the probability of an IBD segment of length $l$ cM persisting for $t$ generations declines quickly as $\exp(-t \times l/50)$), the rate of sporadic sharing of IBD segments probes genealogical connections between groups of individuals only a few hundred years deep, for example, for modern Europeans[2]. To showcase how detecting IBD segments with ancIBD can reveal such connections between ancient individuals, we applied our method to a set of previously published ancient West Eurasian aDNA data dating to the Late Eneolithic and Early Bronze Age (Supplementary Table 3). This period, from 3,000 to 2,000 BCE, was characterized by major gene flow events, where 'Steppe-related' ancestry had a substantial genetic impact throughout Europe (for example, refs. 30,31), leading to widespread genetic admixtures and population turnover as far west as Britain[32] and Iberia[33]. Applying ancIBD to the relevant published aDNA record of 304 ancient Western Eurasians organized into 24 archaeological groups (Supplementary Table 3), we find several intriguing links. Many of those connections were previously proposed and suggested by admixture tests; however, the sharing of long IBD segments now provides definitive evidence for recent co-ancestry and biological interactions, tethering groups together closely in time.

We found that several nomadic Steppe groups associated with the Yamnaya culture that date to around 3,000 BCE share comparably

large amounts of IBD with each other (Fig. 4). This late Eneolithic to Early Bronze Age culture of pastoral nomads, who inhabited the Western Eurasian Pontic-Caspian Steppe often buried their death in tumuli (Kurgans) and were among the first people to use wagons, are suggested to have had a key role in the early spread of Indo-European languages[34]. Notably, the Yamnaya IBD cluster includes also individuals associated with the contemporaneous Afanasievo culture thousands of kilometres east, an Eneolithic archaeological culture near the Central Asian Altai mountains. This signal of IBD sharing confirms the previous archaeological hypothesis that Afanasievo and Yamnaya are closely linked despite the vast geographic distance from Eastern Europe to Central Asia[34]. A genetic link has already been evident from genomic similarity and Y haplogroups[31,35]; however, the time depth of this connection remained unclear. We now identify IBD signals across all length scales, including several shared IBD segments even longer than 20 cM (Extended Data Fig. 3). Such long IBD links must be recent as recombination ends an IBD segment ~20 cM long on average every five meiosis. This long IBD sharing signal, at the same level as between various Yamnaya groups (Fig. 4), therefore clearly indicates that ancient individuals from Afanasievo contexts descend from people who migrated at most a few generations earlier across vast distances of the Eurasian Steppe.

Increased individual mobility in Eneolithic and Early Bronze Age Eurasian Steppe groups is also reflected in a pair of individuals

associated with the Afanasievo culture that were buried 1,410 km apart, one in present-day Central Mongolia and one in Southern Russia, who share several long IBD segments (Fig. 5a,c). We identified four IBD segments 20–40 cM long, a distinctive signal of close biological relatedness typical of about fifth-degree relatives (Fig. 5c,d). Previous work showed that both individuals have a genetic profile typical for Afanasievo individuals and here this close biological link demonstrates that at least one individual in the chain of relatives between them must have travelled several hundreds of kilometres in their lifetime.

Moreover, there are several intriguing observations regarding individuals associated with the Corded Ware culture, an important archaeological culture that appears across a vast area of Eastern, Central and Northern Europe between 3,000 and 2,400 BCE. Previous aDNA research showed Corded Ware groups to be the first people of these regions to carry high amounts of a distinct ancestry found in Eurasian Steppe pastoralists such as the Yamnaya, admixed with previous Final Neolithic farmer cultures[30,31,36,37]. Using IBD, we find that individuals from diverse Corded Ware cultural groups, including from Sweden (associated with the Battle Axe culture), Russia (Fatyanovo) and East/Central Europe share high amounts of long IBD with each other and also have IBD sharing up to 20 cM with various Yamnaya groups (Fig. 4 and Extended Data Fig. 3a,b,c). We find a distinctive IBD signal with the so-called Globular Amphora culture, in particular from Poland and Ukraine, who were Copper Age (Eneolithic) farmers around 3,000 BCE not yet carrying Steppe-like ancestry[38,39]. This IBD link to Globular Amphora appears for all Corded Ware groups in our analysis, including from as far away as Scandinavia and Russia (Fig. 4), which indicates that individuals related to Globular Amphora contexts from Eastern Europe must have had a major demographic impact early on in the genetic admixtures giving rise to various Corded Ware groups.

## Discussion

We have introduced ancIBD, a method to detect IBD segments optimized for aDNA data. The algorithm follows a long line of work using probabilistic HMMs to screen for IBD segments[40–44]. When compared to other methods to detect IBD (IBIS[23], IBDseq[24], Germline[4], Germline2[43] and hapIBD[6]), ancIBD maintains a balanced performance between precision and recall in the low-coverage regime typical for aDNA data. A recent method KIN[45] fits transitions between IBD states to identify relatives up to the third degree but does not identify sporadic IBD segments which are typical of more distant relatives or are useful for demographic inference.

We optimized the default parameters of ancIBD towards performance on imputed 1240k variants, an SNP set widely used in human aDNA. We also recommend downsampling imputed WGS data to this SNP set because using all common 1000 Genome SNPs only marginally improves performance (Supplementary Note 6). Our benchmarks have demonstrated that ancIBD robustly detects IBD longer than 8 cM, for WGS data down to 0.25× and 1240k data down to 1× average coverage depth on 1,240k SNPs. That WGS data perform better than 1240k data at the same coverage depth on target SNPs is not surprising because WGS data cover the entire genome while 1,240k capture data are depleted for off-target data. But imputation at 1240k sites uses all SNPs in the 1000 Genome dataset, thus providing more off-target data leads to substantially improved imputation quality. We found that WGS data can be imputed at roughly three times lower coverage equally as well as 1240k data (Supplementary Fig. 5), consistent with findings from ref. 19. This observation is relevant for choosing aDNA data generation strategies where IBD segment calling is of interest.

We showcased two main applications for identifying long IBD segments within human aDNA. First, ancIBD reveals biological relatives up to the sixth degree as such pairs distinctively share multiple long IBD segments[25]. Allele sharing-based methods commonly used in aDNA studies[46,47] are generally limited to detecting relatives only up to the third degree because they average over the genome and do

not identify signals due to only a few shared IBD segments that make up only a small part of the genome. However, they can be applied to substantially lower coverage than ancIBD. Similarly, KIN[45] can be applied to lower coverage than ancIBD but is also limited to detecting relatives up to the third degree.

Second, identifying IBD segments with intermediate coverage aDNA data unlocks a powerful way to investigate fine-scale genealogical connections of past human populations. Sharing of long haplotypes establishes bounds on the number of generations separating pairs of individuals, which adds information beyond average single-locus correlation statistics that have been the workhorse of aDNA studies to date. To showcase this potential, we have used ancIBD to generate evidence for the origins of the people culturally associated with the Corded Ware culture. Corded Ware groups of Eastern, Central and Northern Europe were identified to be among the first cultures affected by large-scale gene flows starting 3,000 BCE which spread a distinct ancestry found in pastoralists of the Pontic-Caspian Steppes across Europe[30–32]. Our analysis of long IBD segments reveals that the quarter of Corded Ware Complex ancestry associated with earlier European farmers can be pinpointed to people associated with the Globular Amphora culture of Eastern Europe, who carry no Steppe-like ancestry yet, while the remaining three-quarters must share recent co-ancestry with Yamnaya Steppe pastoralists in the late third millennium BCE. This direct evidence that most Corded Ware ancestry must have genealogical links to people associated with Yamnaya culture spanning on the order of at most a few hundred years is inconsistent with the hypothesis that the Steppe-like ancestry in the Corded Ware primarily reflects an origin in as-of-now unsampled cultures genetically similar to the Yamnaya but related to them only a millennium earlier.

Several extensions could improve ancIBD. Both SNP density in the 1240k and 1000 Genome SNP set varies substantially along the genome[29]. We have found that false positive rate negatively correlates with SNP density (Supplementary Fig. 9) and designed a filter to mask genomic regions with high false positive rates of long IBD (Supplementary Fig. 9). Focusing exclusively on regions of high SNP density could enable one to call IBD with shorter lengths. We also note that we have imputed ancient data using a modern reference haplotype panel, which yields decreasing imputation and phasing performance the older the sample[19,48]. Future efforts to include high-quality ancient genomes into reference haplotype panels or to use modern reference panels substantially larger than 1000 Genomes will probably improve the quality of imputed ancient genomes and thus also boost the performance of ancIBD. We note that ancIBD takes imputed data as input, thus future improvements of imputation software or reference panels can be easily integrated by updating the preprocessing step.

Our algorithm infers the presence of at least one shared IBD segment between two diploid individuals but in practice both pairs or even three or all four haplotypes can be shared. Here, we deliberately kept the model simple to improve robustness and runtime. Importantly, we believe that detecting the presence of one IBD segment alone suffices for most practical applications. Double IBD sharing, often termed IBD2, occurs mostly in full siblings, who on average share half of their genome length in a single IBD and one additional quarter in a double IBD. In this case, the sum of IBD length alone distinguishes full siblings from parent–offspring pairs (who distinctively have their whole genome in IBD) and from second-degree relatives (separate clusters in Extended Data Fig. 4). Beyond full siblings, having overlapping IBD segments on different haplotype pairs only rarely occurs in practice[49]. Only in special cases, such as distinguishing double first cousins from other second-degree relatives, identifying double IBD can be useful. In that case, we recommend directly screening for identical imputed genotypes in IBD segments.

One promising extension is calling IBD segments on X chromosomes. Genetic males have only one copy of it, while females have

two, which causes sex-specific inheritance and recombination patterns (for example, males must have inherited their X chromosomes from their mothers). Therefore, IBD sharing on the X chromosome can provide information about sex-specific relatedness and demography[50]. Our work here focused on the autosomes that make up most of the human genome; however, one can in principle apply ancIBD to imputed female X chromosomes. To call IBD on the X in pairs involving males, one could adapt the state space of ancIBD in a technically straightforward way. Another potential application of IBD segments is to improve the dating of ancient samples by using recombination clocks to tether samples in time. Future work to refine carbon-14 dating, a method widely used for determining the age of human remains, can build upon existing Bayesian methods to incorporate external information into such dates[51–53].

Detecting IBD segments in modern DNA has yielded fine-scale insights into the recent demography of present-day populations, allowing researchers to infer population size dynamics[54,55], genealogical connections between various groups of people[2,43,56] and the geographic scale of individual mobility[3,55]. In principle, such analysis can also be applied to aDNA. It is particularly encouraging that the number of sample pairs that can be screened for IBD segments grows quadratically with the sample size, while the number of ancient genomes used in aDNA studies itself is currently quickly growing[57]. This rapid scaling will provide aDNA researchers with a powerful way to address demographic questions about the human past. We believe that the method to detect IBD in aDNA presented here marks only a first step towards creating the next generation of demographic inference tools, resulting in unprecedented insights into the human past.

## Online content

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

## Methods

### Ethics

No new aDNA data were generated for this study and we only analysed previously published and publicly available aDNA data. Identifying biological kin is a standard analysis in the aDNA field. Permission for aDNA work on the archaeological samples was granted by the respective excavators, archaeologists, curators and museum directors of the sites. These permissions are part of the original publications (listed in Supplementary Table 1).

### The HMM

The ancIBD HMM makes use of the imputed genotype probabilities and phase information output by GLIMPSE and, for each pair of samples, runs a forward-backward algorithm[60] to calculate the posterior probabilities of being in an IBD state at each marker (Fig. 1). These probabilities are then postprocessed to call IBD segments. In the following sections, we describe this HMM (Fig. 1a) in detail, in particular its states, the model for emission and transition probabilities, the calling of IBD segments and postprocessing and its implementation.

Throughout, we assume biallelic variants and denote the two individuals we screen for IBD as 1 and 2 and their phased haplotypes as (1A, 1B) and (2A, 2B). The HMM screens each of the 22 autosomal chromosomes from beginning to end independently, thus it suffices to describe the HMM applied to one chromosome.

**Hidden states.** Our HMM has five hidden states $s = 0,1,...,4$. The first state $s = 0$ encodes a non-IBD state, while the four states $s = 1,2,3,4$ encode the four possibilities (1A/2A, 1A/2B, 1B/2A, 1B/2B) of sharing an IBD allele between the haplotypes of two diploid genomes (1A,1B) and (2A,2B) (Fig. 1a). We note that we do not model IBD sharing beyond a single pair of haplotypes (where both pairs of or more than three haplotypes share a recent common ancestor). These cases occur only rarely in practice[49] and our goal here is to identify long tracts of IBD.

**Transition probabilities.** To calculate the $5 \times 5$ transition probabilities $T$ to change states from one to the following loci, denoted by $l$ and $l+1$, we make use of the genetic map distances obtained from a linkage map, that is a map of the position using Morgans as the unit of length (1 M is the genomic map span over which the average number of recombinations in a single generation is 1).

As in ref. 29, we specify the transition probabilities via a $5 \times 5$ infinitesimal transition rate matrix $Q$, from which each transition probability matrix $A_{l \to l+1}$ is obtained through matrix exponentiation using the genetic distance $r_l$ between loci $l$ and $l+1$

$$A_{l \to l+1} = \exp(Q \times r_l).$$

Here, $Q$ is defined by the following three rate parameters: the rate to jump from the non-IBD state into any of the four IBD states (IBDin), the rate to jump from any of the IBD states to the non-IBD states (IBDout) and the rate to jump from any of the IBD states to another one (IBDswitch):

$$Q = \begin{pmatrix} & \text{IBDin} & \text{IBDin} & \text{IBDin} & \text{IBDin} \\ \text{IBDout} & & \text{IBDswitch} & \text{IBDswitch} & \text{IBDswitch} \\ \text{IBDout} & \text{IBDswitch} & & \text{IBDswitch} & \text{IBDswitch} \\ \text{IBDout} & \text{IBDswitch} & \text{IBDswitch} & & \text{IBDswitch} \\ \text{IBDout} & \text{IBDswitch} & \text{IBDswitch} & \text{IBDswitch} & \end{pmatrix}, \quad (1)$$

where the diagonal elements are defined as $Q_{ii} = -\sum_{j \neq i} Q_{ij}$ such that the rows of $Q$ sum to zero as required for a transition rate matrix. The rate IBDswitch models phasing errors, as a transition from one IBD state to another means that a different haplotype pair is shared. We note that the probability of the IBD state jumping from 1A/2A to 1B/2B would require phase switch errors to occur in both individuals at the same genomic location, which is highly unlikely; however, we set the transition matrix between all four IBD states symmetric as this allowed us to implement a substantial computational speed up.

### Emission probabilities

**Single-locus emission probabilities.** To define the emission model of the HMM, we need to specify $P(D|s)$, the likelihood of the genetic data for the five HMM states $s = 0,1,...,4$ at one locus. Throughout, we denote reference and alternative alleles as 0 and 1, respectively, and the corresponding genotype as $g \in \{0,1\}$. The observed data $D$ of our emission model will be the haploid dosage, which is the probability of a phased haplotype carrying an alternative allele, here denoted for each haplotype $h$ as

$$x_h = P(g_h = 1), \quad h \in \{1A, 1B, 2A, 2B\}.$$

First, we explain how we approximate the two haploid dosages for a single imputed diploid individual 1. We have to use an approximation as GLIMPSE only outputs the most likely phased diploid genotype GT $\in \{0|0, 0|1, 1|0, 1|1\}$ as well as three posterior genotype probabilities GP for each of the unphased diploid genotypes, denoted by the number of alternative alleles as 0,1,2. We first approximate the posterior probabilities for the four phased states, here denoted as $P_{00}, P_{01}, P_{10}$ and $P_{11}$. The two homozygote probabilities $P_{00}$ and $P_{11}$ are obtained trivially from the corresponding unphased genotype probabilities GP, as no phase information is required for homozygotes. To obtain probabilities of the two phased heterozygotes states, $P_{01}$ and $P_{10}$, we use a simple approximation. Let $p_0, p_1, p_2$ denote the posterior probability for each of the three possible diploid genotypes. If the maximum-likelihood unphased genotype is heterozygote, that is $\max(p_0, p_1, p_2) = p_1$, we set $P_{01} = p_1, P_{10} = 0$ if GT $= 0|1$ and $P_{01} = 0, P_{10} = p_1$ if GT $= 1|0$. If the maximum-likelihood unphased genotype is a homozygote, that is $\max(p_0, p_1, p_2) = p_0$ or $p_2$ and thus there is no phase information for the heterozygote genotype available, we set $P_{01} = P_{10} = p_1/2$. Having obtained the four probabilities for the possible phased genotypes, we can calculate the two haploid dosages as:

$$x_{1A} = P_{11} + P_{10} \quad (2)$$

$$x_{1B} = P_{11} + P_{01}. \quad (3)$$

When calling IBD segments between two individuals 1 and 2, we use this approach to obtain all four haploid dosages and denote them for haplotypes 1A, 1B, 2A, 2B as $(x_{1A}, x_{1B}, x_{2A}, x_{2B})$.

Setting those four haploid dosages as the observed data $D = (x_{1A}, x_{1B}, x_{2A}, x_{2B})$ at one locus, we can now calculate the likelihood $P(D|s)$ for each of the five HMM states $s = 0,1,...,4$. We start by summing over all possible unobserved latent phased genotypes $\mathbf{g} = (g_{1A}, g_{1B}, g_{2A}, g_{2B})$, yielding in total 16 possible combinations of reference and alternative alleles, denoted together as $\mathcal{G} = \{0,1\} \times \{0,1\} \times \{0,1\} \times \{0,1\}$:

$$P(D|s = i) = \sum_{\mathbf{g} \in \mathcal{G}} P(D|\mathbf{g})P(\mathbf{g}|s = i). \quad (4)$$

For the term $P(D|\mathbf{g})$, applying Bayes rule yields:

$$P(D|\mathbf{g}) = \frac{P(\mathbf{g}|D) \times P(D)}{P(\mathbf{g})}.$$

$P(D)$ remains a constant factor across all states, which can be ignored because posterior probabilities of an HMM remain invariant to constant factors in the likelihood. We arrive at:

$$P(D|s = i) \approx \sum_{\mathbf{g} \in \mathcal{G}} \frac{P(\mathbf{g}|D)}{P(\mathbf{g})} P(\mathbf{g}|s = i). \quad (5)$$

We now approximate the three quantities on the right-hand side of equation (5) for a given set of genotypes **g**.

First, assuming Hardy–Weinberg equilibrium, $P(\mathbf{g})$ is calculated as the product of the four corresponding allele frequencies of (either $p$ or $1-p$ depending on the respective allele in **g** being 0 or 1). In practice, we obtain $p$ from the allele frequencies in the reference panel.

Second, we approximate $P(\mathbf{g}|D)$ as the product of the four probabilities of each of the haplotypes (1A,1B) and (2A,2B) being reference or alternative. We assume that diploid genotype probabilities can be approximated as products of the respective haploid dosages, which we empirically verified on GLIMPSE imputed data (Supplementary Fig. 20). Using the haploid dosages $(x_{1A}, x_{1B}, x_{2A}, x_{2B})$ as calculated above yields:

$$P(\mathbf{g}|D) = \prod_{j \in \{1A,1B,2A,2B\}} [\mathbf{g}_j x_j + (1 - \mathbf{g}_j)(1 - x_j)]. \qquad (6)$$

Third, to approximate $P(\mathbf{g}|s=i)$ we again assume Hardy–Weinberg probabilities which yield a product of factors $p$ or $1-p$ (listed in Supplementary Note 1). For the four IBD states, the two shared alleles constitute one shared draw. Consequently, there are only three instead of four independent factors and genotype combinations **g** where the shared genotype would be different have 0 probability.

Plugging these three approximations into equation (5) now gives $P(D|s)$ for each state $s = 0,1,...,4$.

For the background state ($s = 0$) we have $P(g) = P(g|s=0)$ and thus these factors cancel out in equation (5). Using that $\sum_{\mathbf{g}} P(\mathbf{g}|D) = 1$, we arrive at:

$$P(D|s = 0) = 1. \qquad (7)$$

The four IBD states ($s = 1,2,3,4$) are calculated analogously with a simple rearrangement of the haplotype order. Thus, it suffices to describe $s = 1$, the state where the two first phased genotypes, 1A and 2A, are identical. For the two nonshared alleles the Hardy–Weinberg factors cancel out as in $s = 0$. After some rearranging (Supplementary Note 1), we obtain:

$$P(D|s = 1) = \frac{1}{p} x_{1A} x_{2A} + \frac{1}{1-p}(1 - x_{1A})(1 - x_{2A}). \qquad (8)$$

### Postprocessing: calling IBD segments

To call IBD segments, we use the posterior probability of being in the IBD states obtained via the standard HMM forward-backward algorithm[20], which takes as input the transition rates (equation (1)) and emission probabilities (equations (7) and (8)). Our method then screens for consecutive markers where the posterior probability of being in the non-IBD state $h = 0$ remains below a prespecified threshold. We determine the start of an inferred IBD segment by locating the first SNP whose posterior decreases below the threshold and the end by the first SNP whose posterior rises above the threshold. For each such genomic region longer than a prespecified minimum length cutoff, one IBD segment is recorded.

A postprocessing step commonly applied when detecting IBD is to merge two closely neighbouring IBD segments[2,5]. This step aims to remove spurious gaps within one true IBD segment, which can appear to be caused by low density of SNPs or sporadic genotyping errors. The rationale is that, under most demographic scenarios, sharing of long IBD is very rare and thus two IBD segments are unlikely to occur next to each other by chance[49]. Removing artificial gaps is important for determining the length of an IBD segment and therefore in particular for downstream methods that use the lengths of IBD segments as a recombination clock. In our implementation, we merge all gaps where both IBD are longer than a threshold length and separated by a gap of a maximum length.

By examining rates of IBD segments across the genome when inferring IBD in a large set of empirical aDNA data, we observed excessive rates of IBD sharing in genomic regions with very low SNP density.

This signal is probably driven by false positive IBD segments. We found that filtering IBD segments with an average SNP density of 1240k SNPs below 220 per centimorgan largely attenuates this signal. Additionally, we designed a set of genomic masks to filter 13 regions with generally high levels of IBD sharing (Supplementary Note 5 and Supplementary Fig. 9) that cover about 8% of the genome, with most masked regions involving centromeres and telomeres. The human-specific masking is optional, the SNP density filter is applied by default by ancIBD.

### Setting default parameters of ancIBD

In the following, we describe how we chose the default parameters of ancIBD. In principle, users can specify any SNP set as input but our goal was to obtain default parameters that are optimized for imputed genotype likelihoods at the 1240k SNP set, as most published human aDNA data consists of in-solution DNA capture experiments enriching for this SNP set.

First, we simulated a dataset including ground-truth IBD sharing by using haplotypes in the 1000 Genome Project panel[17]. We simulated chromosome 3 by stitching together short haplotypes 0.25 cM long copied from reference individuals labelled as TSI (Tuscany, Italy) and then copied IBD segments of various lengths (4, 8, 12, 16 and 20 cM) into 100 pairs of mosaic genomes (described in detail in Supplementary Note 2 and Extended Data Fig. 1). This approach, following ref. 2, yields a set of diploid genotype data with exactly known IBD. Such a haplotype mosaic removes long IBD segments in the 1000 Genome data while also maintaining most of the local haplotype structure. To obtain data typical for aDNA sequencing, we matched genotyping errors and probabilities observed within downsampled high-coverage empirical aDNA data and added phase switch errors (Supplementary Note 2).

We then applied ancIBD for a range of parameter combinations and recorded performance statistics (Supplementary Tables 4 and 5). The final parameters that we set as default values (listed in Extended Data Table 3) are chosen to work well for a broad range of coverages and IBD lengths. Throughout this work, we use these settings but, in our implementation, each parameter can be changed to a nondefault value by the user.

### Implementation and runtime

We implemented several computational speed-ups to improve the runtime of our algorithm. First, the forward-backward algorithm is coded in the Cython module to make use of the increased speed of a precompiled C function within our overall Python implementation. Second, our algorithm uses a rescaled version of the forward-backward algorithm[20] which avoids computing logarithms of sums that would be computationally substantially more expensive than products and additions. Finally, we make use of the symmetry of the four IBD states. As the transition probabilities between those are fully symmetric, we can reduce the transition matrix from a 5 × 5 to a 3 × 3 matrix by collapsing the three other IBD states into a single 'other IBD' state. After the exponentiation of the 3 × 3 matrix, the original 5 × 5 transition matrix is reconstructed by dividing up the jump rates using the original symmetry.

We use the Python package scikit-allel (v.1.2.1) to transform the VCF output of GLIMPSE to an HDF5 file, a data format that allows efficient partial access to data[61], for example we can effectively load data for any subset of individuals.

The average runtime of ancIBD (v.0.5) for a pair of imputed individuals on all 22 autosomes is about 25 s when using a single Intel Xeon E5-2697 v.3 CPU with 2.60 GHz (Extended Data Fig. 5). As the number of pairs in a sample of $n$ individuals grows as $n(n-1)/2$, the runtime scales quadratically when screening all pairs of samples for IBD (Extended Data Fig. 5). However, we note that due to the speed of a HMM forward-backward algorithm with five states requiring only a few multiplications and additions per locus, a large fraction of runtime per pair is due to loading the data (Extended Data Fig. 5). Thus, an efficient strategy is to load a set of individuals into memory jointly, as then the

loading time scales only linearly with the number of samples. This strategy, implemented in ancIBD, leads to hugely improved runtime per pair of samples in cases where many samples are loaded into memory and screened for pairwise IBD (Extended Data Fig. 5). We observed that for batches of size 50 samples and when screening all $50 \times 49/2 = 1{,}225$ pairs for IBD, the average runtime of ancIBD per imputed pair for all 22 chromosomes reduces to ~0.75 s. The asymptotic limit per sample pair, which is the runtime of the HMM and postprocessing, is about 0.35 s on our architecture.

## Empirical data analysis

We applied ancIBD to a large set of previously published aDNA data of ancient Eurasians (using the bioinformatic processing described in the AADR dataset[57]). After filtering to all individuals with geographic coordinates in Eurasia dating within the last 45,000 years and sufficient genomic coverage for robust IBD calling we obtained a final set of 4,248 unique ancient individuals (Supplementary Table 1). As the coverage cutoff, we required at least 70% of the 1240k SNPs on chromosome 3 having max(GP) (defined as the maximum among the three posterior genotype probabilities of 0/0,0/1,1/1) exceeding 0.99. This metric was chosen because it can be easily calculated on imputed data for various data types. It corresponds to the coverage cutoff for ancIBD described above, as the relationship between coverage and this metric is monotonic (Supplementary Fig. 19). Our imputation pipeline is described in detail in Supplementary Note 3. We then screened each of the 9,020,628 pairs of ancient genomes with ancIBD. To optimize runtime we grouped the genomes into batches of 400 and then ran all possible pairs between two batches after loading the two batches into memory (this approach is implemented in the in ancIBD software package). For each pair with detected IBD, we collected IBD statistics into a summary table (see Supplementary Table 2 for pairs of published individuals).

## Statistics and reproducibility

For empirical aDNA data analysis presented in this work, we used 4,248 published samples originating from Eurasia dated within the last 45,000 years and passing the coverage requirement. No statistical method was used to predetermine the sample size. All simulation experiments depending on probabilistic random draws were performed with many independent replicates to analyse statistical uncertainty.

## Reporting summary

Further information on research design is available in the Nature Portfolio Reporting Summary linked to this article.

## Data availability

No new DNA data were generated for this study. The reference panel data that we used for imputation (phased haplotypes from the 1000 Genomes dataset) are publicly available at http://ftp.1000genomes. ebi.ac.uk/vol1/ftp/release/20130502/. The four high-coverage genomes used in empirical downsampling experiments were previously published[21] and are available at https://reich.hms.harvard. edu/ancient-genome-diversity-project. The Hazleton samples can be downloaded through the European Nucleotide Archive under accession PRJEB46958. Raw sequencing data of the published West Eurasian ancient individuals are publicly available as described in the original publications (Supplementary Table 1). The AADR resource including the metadata we used are publicly available at https://reich.hms. harvard.edu/allen-ancient-dna-resource-aadr-downloadable-genotypes-present-day-and-ancient-dna-data. We deposited a table of all inferred IBD segments between the 4,248 ancient individuals at https:// zenodo.org/record/8417049. Source data are provided with this paper.

## Code availability

A Python package implementing the method is available on the Python Package Index (https://pypi.org/project/ancIBD/) and can be installed through pip. Online documentation is available at https://ancibd. readthedocs.io/en/latest/index.html. Code developed for simulating data, analysis and data visualization presented in this study is available at the GitHub repository https://github.com/hringbauer/ancIBD. External softwares used in this study were obtained as follows: bcftools (1.14-26-g018607e), https://samtools.github.io/bcftools/; samtools (v.1.13), http://www.htslib.org/; GLIMPSE (v.1.1.1), https://odelaneau. github.io/GLIMPSE/glimpse1/; ibis (v.1.20.9), https://github.com/ williamslab/ibis; ped-sim (v1.4), https://github.com/williamslab/ped-sim; IBDseq (r1206), https://faculty.washington.edu/browning/ibdseq.html; hapIBD (v.1.0, 1.0, 23Apr20.f1a), https://github.com/browning-lab/ hap-ibd; GERMLINE2 (v.1.0), https://github.com/gusevlab/germline2; GERMLINE (1.5.3), http://gusevlab.org/projects/germline/; scikit-allel (v.1.2.1), https://pypi.org/project/scikit-allel/; Cython (v.0.29.14), https://pypi.org/project/Cython/.

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

## Acknowledgements

We thank S. Carmi (Hebrew University of Jerusalem) for insightful comments on this paper. We gratefully acknowledge useful discussions with members of the Reich laboratory (Harvard University) and with the population genetics meeting group at the MPI-EVA Leipzig. We thank M. de Brito for her useful feedback. This work was supported by the National Institutes of Health grant HG012287 (D.R.), by the John Templeton Foundation grant 61220 (D.R.), by the Howard Hughes Medical Institute (D.R.) and by funding from the Max Planck Society (H.R.). The funders had no role in study design, data collection, analysis, decision to publish or preparation of the manuscript.

## Author contributions

H.R., D.R. and N.P. designed this study. H.R. and Y.H. developed the software. H.R., Y.H., A.A., I.O. and S.M. conducted the formal analysis. A.A., D.R., H.R., S.M. and I.O. were responsible for data curation. D.R. and N.P. undertook supervision. D.R. was responsible for funding acquisition. H.R. and Y.H. created the visualization and wrote the original paper. All authors were involved in reviewing and editing the final paper.

## Funding

## Competing interests

The authors declare no competing interests.

## Additional information

**Extended data** is available for this paper at https://doi.org/10.1038/s41588-023-01582-w.

**Correspondence and requests for materials** should be addressed to Harald Ringbauer or David Reich.

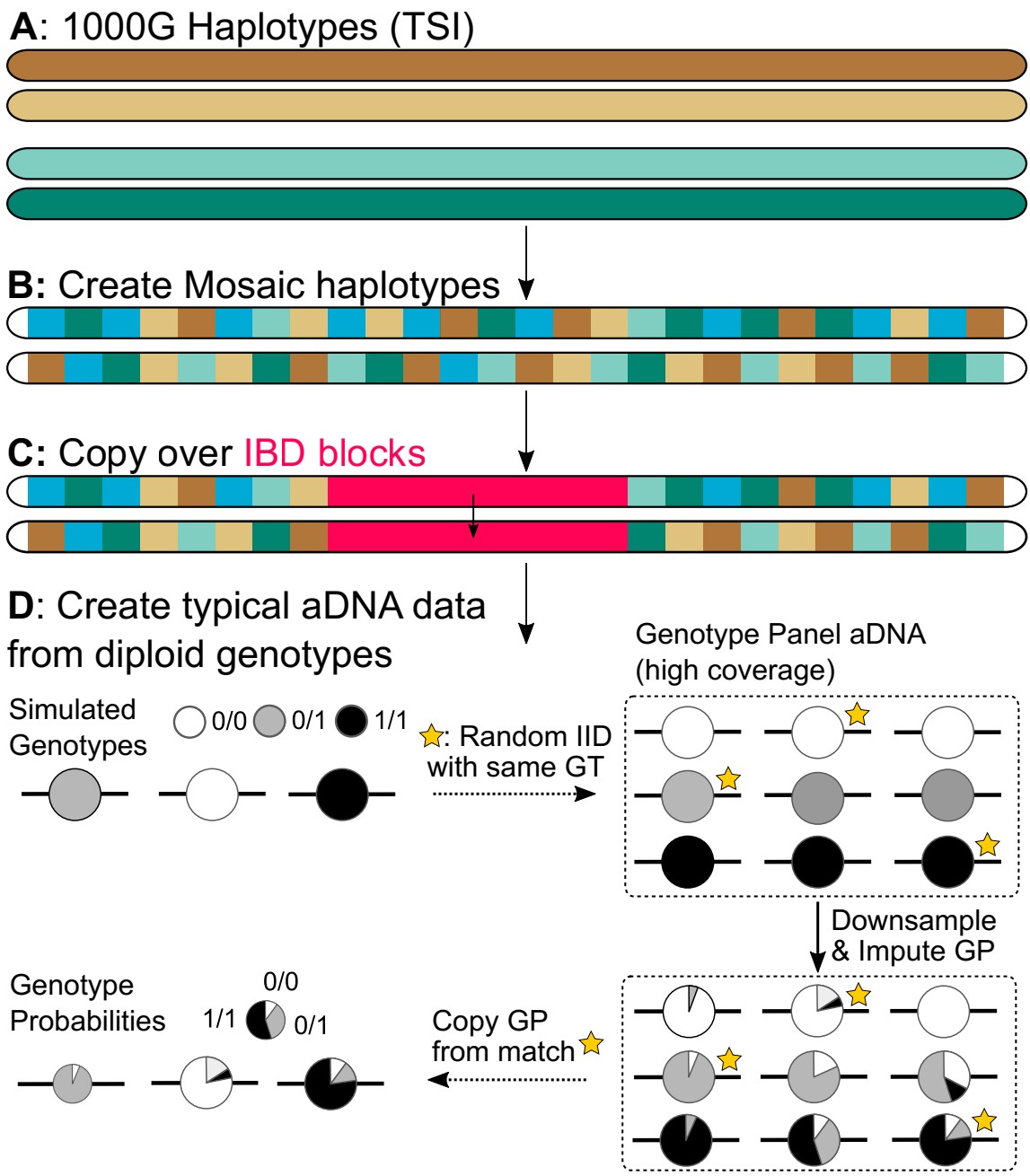

**Extended Data Fig. 1 | Pipeline to simulate IBD segment data.** We visualize our steps to simulate IBD segment data (see detailed description in Supplementary Note 2). Starting from TSI (Tuscany) high-quality reference haplotypes in the 1000 Genome panel (**A**), we created haplotype mosaics (**B**) as any long IBD segment is removed from those. We then copied over IBD segments of the target length (**C**). We grouped two mosaic haplotypes to obtain diploid individuals but to simplify visualization here we do not depict the second haplotype per individual. (**D**): To create data typical for imputed low-coverage aDNA, we matched each genotype to a random matching genotype in a panel of aDNA diploid genotypes called from high-coverage aDNA (either 1240k or WGS aDNA data). We then downsampled the high-coverage aDNA panel to the target coverage, imputed genotype probabilities and copied those back to each match.

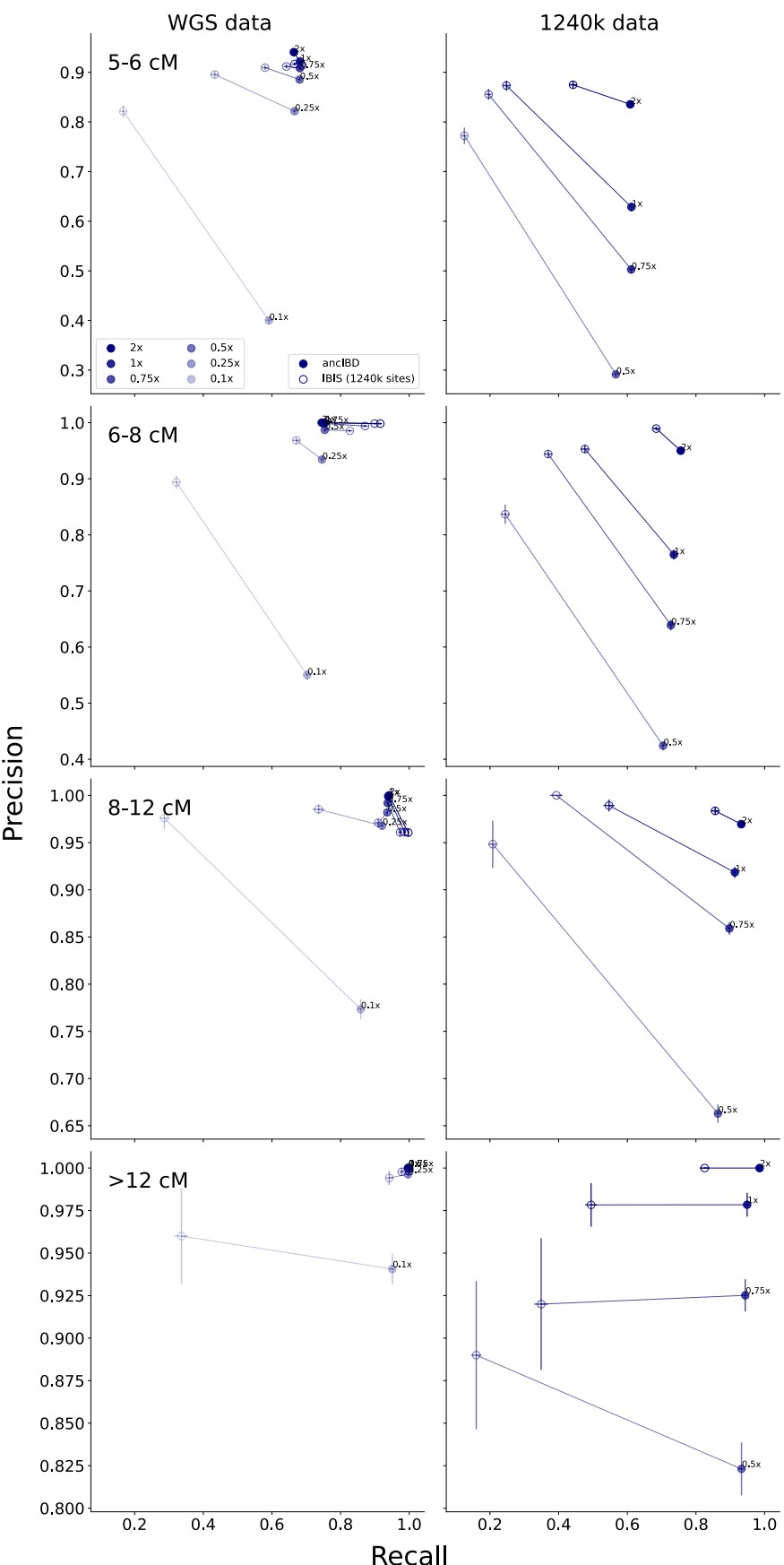

**Extended Data Fig. 2 | Precision and recall of ancIBD and IBIS at various length bins and coverages.** We applied both methods with their default settings to genotype data imputed after downsampling to various coverages. For each coverage, we report the average precision and recall of each length bin across 50 independent replicates. The error bar represents ± SE of the estimated precision and recall. Each row represents a length bin and each column represents one input data type (either WGS data or 1240k data). Note that the y axis ranges are different for different rows.

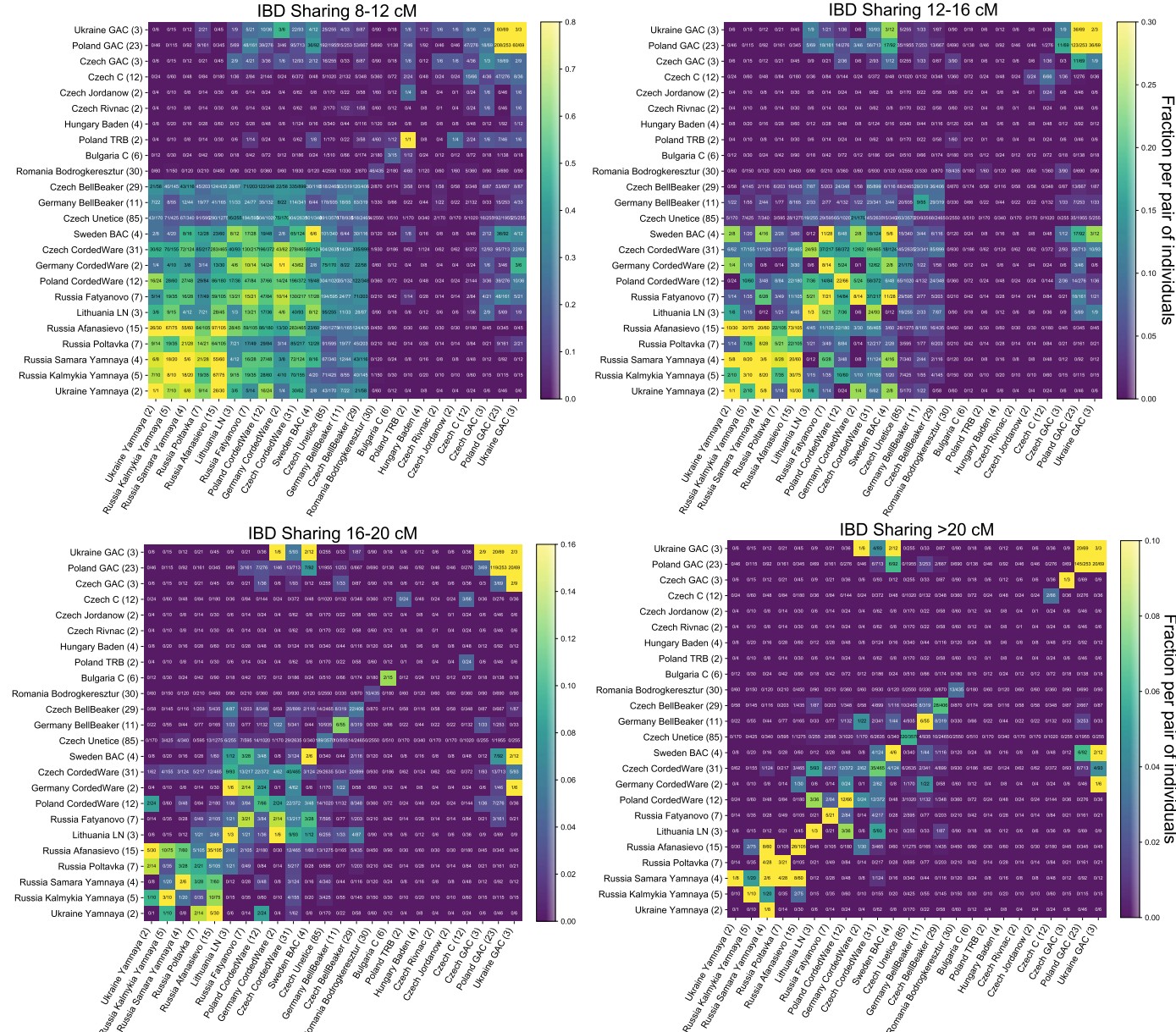

**Extended Data Fig. 3 | IBD sharing matrix of various Eneolithic & Bronze Age West Eurasian Groups for four IBD length scales.** As in Fig. 4, but for shared IBD [8 − 12 cM], [12 − 16 cM], [16 − 20 cM], > 20 cM long. We used ancIBD to infer IBD segments between all pairs of groups and visualize the fraction of pairs that share at least one IBD for each pair of populations and for the four different IBD length bins.

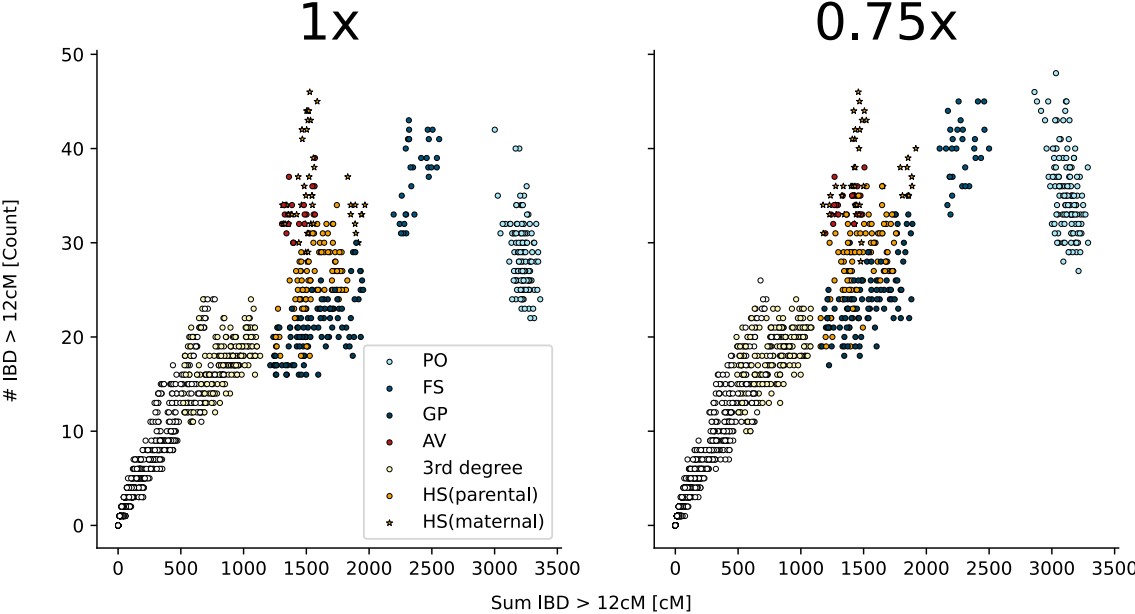

**Extended Data Fig. 4 | Downsampling of Hazelton pedigree samples.** We downsampled all individuals from a previously published English Neolithic pedigree[26] with coverage at least 1x both to 1x and 0.75x. For each coverage, we downsampled 10 times, each with different random seeds, to create 10 replicates. Therefore, not all dots are independent pairs of relatives; they may be the same pair downsampled with different random seeds. The relationship annotations are obtained from Supp. Table 5 of ref. 26. All relatives more distant than 3rd degree are depicted as hollow dots.

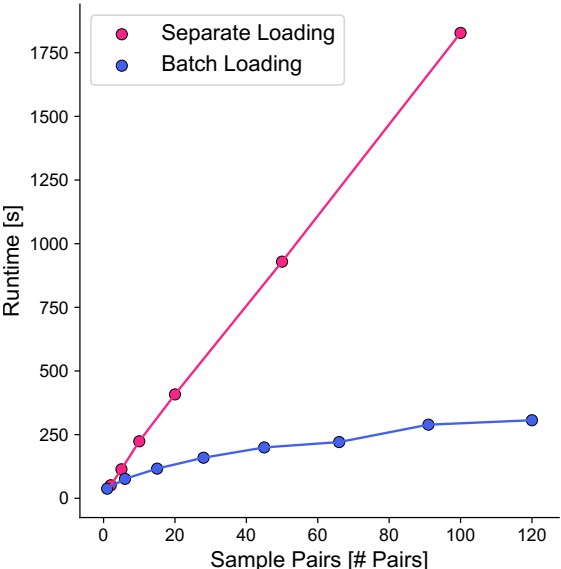

**Extended Data Fig. 5 | Runtime Benchmarks of ancIBD.** To benchmark runtimes, we applied ancIBD on empirical ancient DNA data in .hdf5 format imputed at 1240k sites. We used the imputed hdf5 file from the Eurasian application (Fig. 3), choosing samples and pairs at random. **Left:** For each sample pair, all autosomes are screened for IBD. In one experiment all pairs of samples were run independently, leading to a linear dependency on pair number, as expected. In a second experiment, all samples were loaded into memory and then each sample pair was screened for IBD. The apparent sub-linear behaviour is due to the fact that loading $n$ samples scales slower than the actual runtime of $n(n-1)/2$ sample pairs. **Right:** We depict the runtimes normalized per sample pair when screening all pairs of sample batches of various sizes for IBD. We visualize the loading time (the time it takes to load the hdf5 genotype data into memory), the preprocessing time (including preparing the transition and emission matrix), as well as the runtime of screening for IBD that includes the forward-backward algorithm as well as postprocessing. Due to the decrease in the impact of the time to load the data, which scales linearly with batch size while the number of sample pair scales quadratically, we observe substantially increased runtimes per pair.

**Extended Data Table 1 | Inferred segment length in simulated WGS-like data**

|        | 4cM   | 8cM   | 12cM   | 16cM   | 20cM   |
|--------|-------|-------|--------|--------|--------|
| **2x**    | 4.270 | 8.286 | 12.297 | 16.279 | 20.297 |
| **1x**    | 4.341 | 8.351 | 12.334 | 16.311 | 20.349 |
| **0.75x** | 4.382 | 8.388 | 12.391 | 16.350 | 20.370 |
| **0.5x**  | 4.448 | 8.469 | 12.454 | 16.422 | 20.431 |
| **0.25x** | 4.686 | 8.664 | 12.682 | 16.663 | 20.652 |
| **0.1x**  | 5.182 | 9.657 | 13.755 | 17.957 | 21.606 |

For each of the simulated IBD lengths (4cM, 8cM, 12cM, 16cM, 20cM) with WGS-like data quality at various coverages, the table shows the inferred segment length averaged over 500 independent replicates.

**Extended Data Table 2 | Inferred segment length in simulated 1240k-like data**

|         | 4cM   | 8cM    | 12cM   | 16cM   | 20cM   |
|---------|-------|--------|--------|--------|--------|
| **2x**     | 4.559 | 8.592  | 12.572 | 16.551 | 20.528 |
| **1x**     | 4.825 | 8.961  | 12.956 | 17.000 | 20.860 |
| **0.75x**  | 5.005 | 9.270  | 13.281 | 17.442 | 21.242 |
| **0.5x**   | 5.196 | 10.003 | 14.287 | 18.391 | 22.094 |
| **0.25x**  | 5.699 | 11.526 | 17.112 | 21.209 | 26.062 |
| **0.1x**   | 6.017 | 12.002 | 19.161 | 25.150 | 30.370 |

For each of the simulated IBD lengths (4cM, 8cM, 12cM, 16cM, 20cM) with 1240k-like data quality at various coverages, the table shows the inferred segment length averaged over 500 independent replicates.

**Extended Data Table 3 | Parameters of ancIBD HMM and default values**

| Parameter | Value | Description [Unit] |
|---|---:|---|
| **HMM** | | |
| ibd_in | 1 | Transition rate out of non-IBD state [per Morgan] |
| ibd_out | 10 | Transition rate out of IBD states [per Morgan] |
| ibd_jump | 400 | Transition rate between IBD states [per Morgan] |
| min_error | 0.001 | Cap Min. Probabilitiy of diploid genotype Error [rate] |
| p | Sample/1000G | Allele Frequencies (AF) |
| **Post-Processing** | | |
| cutoff_post | 0.99 | Minimum Posterior on IBD state for IBD segment call |
| min_cm | 2 | Minimum Length IBD segment to call [centimorgan] |
| snp_cm | 220 | Minimum SNP density in IBD [per centimorgan] |
| max_gap | 0.0075 | Maximum Gap to merge [Morgan] |

All parameters that can be set in our implementation. The default values are chosen to work well (low FP, high power, little length bias and variance) for a broad range of WGS and 1240k aDNA data.

# Reporting Summary

## Statistics

For all statistical analyses, confirm that the following items are present in the figure legend, table legend, main text, or Methods section.

| n/a | Confirmed | |
|---|---|---|
| ☐ | ☒ | The exact sample size (*n*) for each experimental group/condition, given as a discrete number and unit of measurement |
| ☐ | ☒ | A statement on whether measurements were taken from distinct samples or whether the same sample was measured repeatedly |
| ☐ | ☒ | The statistical test(s) used AND whether they are one- or two-sided *Only common tests should be described solely by name; describe more complex techniques in the Methods section.* |
| ☒ | ☐ | A description of all covariates tested |
| ☐ | ☒ | A description of any assumptions or corrections, such as tests of normality and adjustment for multiple comparisons |
| ☐ | ☒ | A full description of the statistical parameters including central tendency (e.g. means) or other basic estimates (e.g. regression coefficient) AND variation (e.g. standard deviation) or associated estimates of uncertainty (e.g. confidence intervals) |
| ☒ | ☐ | For null hypothesis testing, the test statistic (e.g. *F*, *t*, *r*) with confidence intervals, effect sizes, degrees of freedom and *P* value noted *Give P values as exact values whenever suitable.* |
| ☐ | ☒ | For Bayesian analysis, information on the choice of priors and Markov chain Monte Carlo settings |
| ☒ | ☐ | For hierarchical and complex designs, identification of the appropriate level for tests and full reporting of outcomes |
| ☐ | ☒ | Estimates of effect sizes (e.g. Cohen's *d*, Pearson's *r*), indicating how they were calculated |

*Our web collection on statistics for biologists contains articles on many of the points above.*

## Software and code

Policy information about availability of computer code

| Data collection | All custom code used to prepare the data is available on https://github.com/hringbauer/ancIBD. To prepare the data we used "bcftools/1.14" to filter SNPs, "GLIMPSE 1.1.1" to impute genotype probabilities, and the Python package "scikit allele/1.2.1" to transform vcf data to hdf5 data that serves as input for our new software. A detailed description of the data preprocessing steps is given in the Methods section of the manuscript. |
|---|---|
| Data analysis | All custom code used to analyze the data is available on https://github.com/hringbauer/ancIBD. Morover, the new software used to analyze the data is available as Python package "ancIBD" (https://pypi.org/project/hapROH/) that can be installed with pip.<br><br>Relevant published software we used (including versions):<br>bcftools 1.14-26-g018607e; samtools 1.13; GLIMPSE 1.1.1; Python 3.7.4, Python packages: scikit-allele (1.2.1)<br>Cython (0.29.14)<br><br>Software to call IBD we used:<br>ancIBD (the new method presented in this manuscript) 0.5;<br>ibis v1.20.9; IBDseq r1206; hapIBD version 1.0; germline2; germline 1.5.3, hapIBD (1.0) |

For manuscripts utilizing custom algorithms or software that are central to the research but not yet described in published literature, software must be made available to editors and reviewers. We strongly encourage code deposition in a community repository (e.g. GitHub). See the Nature Portfolio guidelines for submitting code & software for further information.

## Data

Policy information about availability of data

All manuscripts must include a data availability statement. This statement should provide the following information, where applicable:

- Accession codes, unique identifiers, or web links for publicly available datasets
- A description of any restrictions on data availability
- For clinical datasets or third party data, please ensure that the statement adheres to our policy

No new DNA data were generated for this study. The reference panel data that we used for imputation (phased haplotypes from the 1000 Genomes dataset) are publicly available at http://ftp.1000genomes.ebi.ac.uk/vol1/ftp/release/20130502/. The four high-coverage genomes used in empirical downsampling experiments were previously published (Wohns et al 2022) and are available at https://reich.hms.harvard.edu/ancient-genome-diversity-project. The Hazleton samples can be downloaded through the European Nucleotide Archive under accession PRJEB46958. Raw sequencing data of the published Westeurasian ancient individuals are publicly available as described in the original publications (see Supplementary Table 1). The AADR resource including the metadata we used is publicly available at https://reich.hms.harvard.edu/allen-ancient-dna-resource-aadr-downloadable-genotypes-present-day-and-ancient-dna-data. We deposited the inferred IBD segments between the 4,248 ancient individuals at https://zenodo.org/record/8417049.

## Human research participants

Policy information about studies involving human research participants and Sex and Gender in Research.

| | |
|---|---|
| Reporting on sex and gender | N/A |
| Population characteristics | N/A |
| Recruitment | N/A |
| Ethics oversight | N/A |

Note that full information on the approval of the study protocol must also be provided in the manuscript.

# Field-specific reporting

Please select the one below that is the best fit for your research. If you are not sure, read the appropriate sections before making your selection.

☒ Life sciences    ☐ Behavioural & social sciences    ☐ Ecological, evolutionary & environmental sciences

For a reference copy of the document with all sections, see nature.com/documents/nr-reporting-summary-flat.pdf

# Life sciences study design

All studies must disclose on these points even when the disclosure is negative.

| | |
|---|---|
| Sample size | In the simulation experiments, we set the number of random replicates (on the order of a few hundreds per parameter set) so that there is sufficient power to test the overall performance of the method while remaining computationally feasible on a scientific cluster. In the real-data experiments, the sample set was determined by the number of publicly available ancient DNA samples. |
| Data exclusions | We used all available ancient DNA data of sufficient quality (meeting the coverage cutoff described in our manuscript). |
| Replication | Our algorithm gives deterministic and therefore replicable results (i.e. on the empirical ancient DNA data analyzed in this work). All downsampling and simulation experiments to generate data depending on probabilistic random draws were performed with a large number of independent replicates to analyze statistical uncertainty. |
| Randomization | The random draws for downsampling and simulation experiments where computationally obtained without any intervention of the respective analyst. Each such experiment was replicated a large number of times to acount for statistical uncertainty. |
| Blinding | Our study is purely computational. On empirical and simulated data, our algorithm gives deterministic results - there is no possible bias of the outcome based on the person running the code. |

# Behavioural & social sciences study design

All studies must disclose on these points even when the disclosure is negative.

| | |
|---|---|
| Study description | *Briefly describe the study type including whether data are quantitative, qualitative, or mixed-methods (e.g. qualitative cross-sectional, quantitative experimental, mixed-methods case study).* |
| Research sample | *State the research sample (e.g. Harvard university undergraduates, villagers in rural India) and provide relevant demographic information (e.g. age, sex) and indicate whether the sample is representative. Provide a rationale for the study sample chosen. For studies involving existing datasets, please describe the dataset and source.* |
| Sampling strategy | *Describe the sampling procedure (e.g. random, snowball, stratified, convenience). Describe the statistical methods that were used to predetermine sample size OR if no sample-size calculation was performed, describe how sample sizes were chosen and provide a rationale for why these sample sizes are sufficient. For qualitative data, please indicate whether data saturation was considered, and what criteria were used to decide that no further sampling was needed.* |
| Data collection | *Provide details about the data collection procedure, including the instruments or devices used to record the data (e.g. pen and paper, computer, eye tracker, video or audio equipment) whether anyone was present besides the participant(s) and the researcher, and whether the researcher was blind to experimental condition and/or the study hypothesis during data collection.* |
| Timing | *Indicate the start and stop dates of data collection. If there is a gap between collection periods, state the dates for each sample cohort.* |
| Data exclusions | *If no data were excluded from the analyses, state so OR if data were excluded, provide the exact number of exclusions and the rationale behind them, indicating whether exclusion criteria were pre-established.* |
| Non-participation | *State how many participants dropped out/declined participation and the reason(s) given OR provide response rate OR state that no participants dropped out/declined participation.* |
| Randomization | *If participants were not allocated into experimental groups, state so OR describe how participants were allocated to groups, and if allocation was not random, describe how covariates were controlled.* |

# Ecological, evolutionary & environmental sciences study design

All studies must disclose on these points even when the disclosure is negative.

| | |
|---|---|
| Study description | *Briefly describe the study. For quantitative data include treatment factors and interactions, design structure (e.g. factorial, nested, hierarchical), nature and number of experimental units and replicates.* |
| Research sample | *Describe the research sample (e.g. a group of tagged Passer domesticus, all Stenocereus thurberi within Organ Pipe Cactus National Monument), and provide a rationale for the sample choice. When relevant, describe the organism taxa, source, sex, age range and any manipulations. State what population the sample is meant to represent when applicable. For studies involving existing datasets, describe the data and its source.* |
| Sampling strategy | *Note the sampling procedure. Describe the statistical methods that were used to predetermine sample size OR if no sample-size calculation was performed, describe how sample sizes were chosen and provide a rationale for why these sample sizes are sufficient.* |
| Data collection | *Describe the data collection procedure, including who recorded the data and how.* |
| Timing and spatial scale | *Indicate the start and stop dates of data collection, noting the frequency and periodicity of sampling and providing a rationale for these choices. If there is a gap between collection periods, state the dates for each sample cohort. Specify the spatial scale from which the data are taken* |
| Data exclusions | *If no data were excluded from the analyses, state so OR if data were excluded, describe the exclusions and the rationale behind them, indicating whether exclusion criteria were pre-established.* |
| Reproducibility | *Describe the measures taken to verify the reproducibility of experimental findings. For each experiment, note whether any attempts to repeat the experiment failed OR state that all attempts to repeat the experiment were successful.* |
| Randomization | *Describe how samples/organisms/participants were allocated into groups. If allocation was not random, describe how covariates were controlled. If this is not relevant to your study, explain why.* |
| Blinding | *Describe the extent of blinding used during data acquisition and analysis. If blinding was not possible, describe why OR explain why blinding was not relevant to your study.* |

Did the study involve field work? ☐ Yes ☐ No

# Field work, collection and transport

| | |
|---|---|
| Field conditions | *Describe the study conditions for field work, providing relevant parameters (e.g. temperature, rainfall).* |
| Location | *State the location of the sampling or experiment, providing relevant parameters (e.g. latitude and longitude, elevation, water depth).* |
| Access & import/export | *Describe the efforts you have made to access habitats and to collect and import/export your samples in a responsible manner and in compliance with local, national and international laws, noting any permits that were obtained (give the name of the issuing authority, the date of issue, and any identifying information).* |
| Disturbance | *Describe any disturbance caused by the study and how it was minimized.* |

# Reporting for specific materials, systems and methods

We require information from authors about some types of materials, experimental systems and methods used in many studies. Here, indicate whether each material, system or method listed is relevant to your study. If you are not sure if a list item applies to your research, read the appropriate section before selecting a response.

### Materials & experimental systems

| n/a | Involved in the study |
|---|---|
| ☒ | ☐ Antibodies |
| ☒ | ☐ Eukaryotic cell lines |
| ☐ | ☒ Palaeontology and archaeology |
| ☒ | ☐ Animals and other organisms |
| ☒ | ☐ Clinical data |
| ☒ | ☐ Dual use research of concern |

### Methods

| n/a | Involved in the study |
|---|---|
| ☒ | ☐ ChIP-seq |
| ☒ | ☐ Flow cytometry |
| ☒ | ☐ MRI-based neuroimaging |

## Antibodies

| | |
|---|---|
| Antibodies used | *Describe all antibodies used in the study; as applicable, provide supplier name, catalog number, clone name, and lot number.* |
| Validation | *Describe the validation of each primary antibody for the species and application, noting any validation statements on the manufacturer's website, relevant citations, antibody profiles in online databases, or data provided in the manuscript.* |

## Eukaryotic cell lines

Policy information about cell lines and Sex and Gender in Research

| | |
|---|---|
| Cell line source(s) | *State the source of each cell line used and the sex of all primary cell lines and cells derived from human participants or vertebrate models.* |
| Authentication | *Describe the authentication procedures for each cell line used OR declare that none of the cell lines used were authenticated.* |
| Mycoplasma contamination | *Confirm that all cell lines tested negative for mycoplasma contamination OR describe the results of the testing for mycoplasma contamination OR declare that the cell lines were not tested for mycoplasma contamination.* |
| Commonly misidentified lines (See ICLAC register) | *Name any commonly misidentified cell lines used in the study and provide a rationale for their use.* |

## Palaeontology and Archaeology

| | |
|---|---|
| Specimen provenance | We did not generate any new ancient DNA data for this study. |
| Specimen deposition | n/a |
| Dating methods | We used the dates from the compiled dataset, which are a mix of the radiocarbon dates and context dates. For details, see each original publications. |

☐ Tick this box to confirm that the raw and calibrated dates are available in the paper or in Supplementary Information.

| Ethics oversight | No ethical approval/guidance was required as this study does not produce any new ancient DNA data. |

Note that full information on the approval of the study protocol must also be provided in the manuscript.

# Animals and other research organisms

Policy information about studies involving animals; ARRIVE guidelines recommended for reporting animal research, and Sex and Gender in Research

| Laboratory animals | *For laboratory animals, report species, strain and age OR state that the study did not involve laboratory animals.* |
| Wild animals | *Provide details on animals observed in or captured in the field; report species and age where possible. Describe how animals were caught and transported and what happened to captive animals after the study (if killed, explain why and describe method; if released, say where and when) OR state that the study did not involve wild animals.* |
| Reporting on sex | *Indicate if findings apply to only one sex; describe whether sex was considered in study design, methods used for assigning sex. Provide data disaggregated for sex where this information has been collected in the source data as appropriate; provide overall numbers in this Reporting Summary. Please state if this information has not been collected. Report sex-based analyses where performed, justify reasons for lack of sex-based analysis.* |
| Field-collected samples | *For laboratory work with field-collected samples, describe all relevant parameters such as housing, maintenance, temperature, photoperiod and end-of-experiment protocol OR state that the study did not involve samples collected from the field.* |
| Ethics oversight | *Identify the organization(s) that approved or provided guidance on the study protocol, OR state that no ethical approval or guidance was required and explain why not.* |

Note that full information on the approval of the study protocol must also be provided in the manuscript.

# Clinical data

Policy information about clinical studies
All manuscripts should comply with the ICMJE guidelines for publication of clinical research and a completed CONSORT checklist must be included with all submissions.

| Clinical trial registration | *Provide the trial registration number from ClinicalTrials.gov or an equivalent agency.* |
| Study protocol | *Note where the full trial protocol can be accessed OR if not available, explain why.* |
| Data collection | *Describe the settings and locales of data collection, noting the time periods of recruitment and data collection.* |
| Outcomes | *Describe how you pre-defined primary and secondary outcome measures and how you assessed these measures.* |

# Dual use research of concern

Policy information about dual use research of concern

## Hazards

Could the accidental, deliberate or reckless misuse of agents or technologies generated in the work, or the application of information presented in the manuscript, pose a threat to:

No | Yes
☐ ☐ Public health
☐ ☐ National security
☐ ☐ Crops and/or livestock
☐ ☐ Ecosystems
☐ ☐ Any other significant area

## Experiments of concern

Does the work involve any of these experiments of concern:

No | Yes
☐ ☐ Demonstrate how to render a vaccine ineffective
☐ ☐ Confer resistance to therapeutically useful antibiotics or antiviral agents
☐ ☐ Enhance the virulence of a pathogen or render a nonpathogen virulent
☐ ☐ Increase transmissibility of a pathogen
☐ ☐ Alter the host range of a pathogen
☐ ☐ Enable evasion of diagnostic/detection modalities
☐ ☐ Enable the weaponization of a biological agent or toxin
☐ ☐ Any other potentially harmful combination of experiments and agents

# ChIP-seq

## Data deposition

☐ Confirm that both raw and final processed data have been deposited in a public database such as GEO.

☐ Confirm that you have deposited or provided access to graph files (e.g. BED files) for the called peaks.

| | |
|---|---|
| Data access links *May remain private before publication.* | *For "Initial submission" or "Revised version" documents, provide reviewer access links. For your "Final submission" document, provide a link to the deposited data.* |
| Files in database submission | *Provide a list of all files available in the database submission.* |
| Genome browser session (e.g. UCSC) | *Provide a link to an anonymized genome browser session for "Initial submission" and "Revised version" documents only, to enable peer review. Write "no longer applicable" for "Final submission" documents.* |

## Methodology

| | |
|---|---|
| Replicates | *Describe the experimental replicates, specifying number, type and replicate agreement.* |
| Sequencing depth | *Describe the sequencing depth for each experiment, providing the total number of reads, uniquely mapped reads, length of reads and whether they were paired- or single-end.* |
| Antibodies | *Describe the antibodies used for the ChIP-seq experiments; as applicable, provide supplier name, catalog number, clone name, and lot number.* |
| Peak calling parameters | *Specify the command line program and parameters used for read mapping and peak calling, including the ChIP, control and index files used.* |
| Data quality | *Describe the methods used to ensure data quality in full detail, including how many peaks are at FDR 5% and above 5-fold enrichment.* |
| Software | *Describe the software used to collect and analyze the ChIP-seq data. For custom code that has been deposited into a community repository, provide accession details.* |

# Flow Cytometry

## Plots

Confirm that:

☐ The axis labels state the marker and fluorochrome used (e.g. CD4-FITC).

☐ The axis scales are clearly visible. Include numbers along axes only for bottom left plot of group (a 'group' is an analysis of identical markers).

☐ All plots are contour plots with outliers or pseudocolor plots.

☐ A numerical value for number of cells or percentage (with statistics) is provided.

## Methodology

| | |
|---|---|
| Sample preparation | *Describe the sample preparation, detailing the biological source of the cells and any tissue processing steps used.* |
| Instrument | *Identify the instrument used for data collection, specifying make and model number.* |

| Software | *Describe the software used to collect and analyze the flow cytometry data. For custom code that has been deposited into a community repository, provide accession details.* |
| --- | --- |
| Cell population abundance | *Describe the abundance of the relevant cell populations within post-sort fractions, providing details on the purity of the samples and how it was determined.* |
| Gating strategy | *Describe the gating strategy used for all relevant experiments, specifying the preliminary FSC/SSC gates of the starting cell population, indicating where boundaries between "positive" and "negative" staining cell populations are defined.* |

☐ Tick this box to confirm that a figure exemplifying the gating strategy is provided in the Supplementary Information.

# Magnetic resonance imaging

## Experimental design

| Design type | *Indicate task or resting state; event-related or block design.* |
| --- | --- |
| Design specifications | *Specify the number of blocks, trials or experimental units per session and/or subject, and specify the length of each trial or block (if trials are blocked) and interval between trials.* |
| Behavioral performance measures | *State number and/or type of variables recorded (e.g. correct button press, response time) and what statistics were used to establish that the subjects were performing the task as expected (e.g. mean, range, and/or standard deviation across subjects).* |

## Acquisition

| Imaging type(s) | *Specify: functional, structural, diffusion, perfusion.* |
| --- | --- |
| Field strength | *Specify in Tesla* |
| Sequence & imaging parameters | *Specify the pulse sequence type (gradient echo, spin echo, etc.), imaging type (EPI, spiral, etc.), field of view, matrix size, slice thickness, orientation and TE/TR/flip angle.* |
| Area of acquisition | *State whether a whole brain scan was used OR define the area of acquisition, describing how the region was determined.* |

Diffusion MRI ☐ Used ☐ Not used

## Preprocessing

| Preprocessing software | *Provide detail on software version and revision number and on specific parameters (model/functions, brain extraction, segmentation, smoothing kernel size, etc.).* |
| --- | --- |
| Normalization | *If data were normalized/standardized, describe the approach(es): specify linear or non-linear and define image types used for transformation OR indicate that data were not normalized and explain rationale for lack of normalization.* |
| Normalization template | *Describe the template used for normalization/transformation, specifying subject space or group standardized space (e.g. original Talairach, MNI305, ICBM152) OR indicate that the data were not normalized.* |
| Noise and artifact removal | *Describe your procedure(s) for artifact and structured noise removal, specifying motion parameters, tissue signals and physiological signals (heart rate, respiration).* |
| Volume censoring | *Define your software and/or method and criteria for volume censoring, and state the extent of such censoring.* |

## Statistical modeling & inference

| Model type and settings | *Specify type (mass univariate, multivariate, RSA, predictive, etc.) and describe essential details of the model at the first and second levels (e.g. fixed, random or mixed effects; drift or auto-correlation).* |
| --- | --- |
| Effect(s) tested | *Define precise effect in terms of the task or stimulus conditions instead of psychological concepts and indicate whether ANOVA or factorial designs were used.* |

Specify type of analysis: ☐ Whole brain ☐ ROI-based ☐ Both

| Statistic type for inference (See Eklund et al. 2016) | *Specify voxel-wise or cluster-wise and report all relevant parameters for cluster-wise methods.* |
| --- | --- |
| Correction | *Describe the type of correction and how it is obtained for multiple comparisons (e.g. FWE, FDR, permutation or Monte Carlo).* |

## Models & analysis

Functional and/or effective connectivity

*Report the measures of dependence used and the model details (e.g. Pearson correlation, partial correlation, mutual information).*

Graph analysis

*Report the dependent variable and connectivity measure, specifying weighted graph or binarized graph, subject- or group-level, and the global and/or node summaries used (e.g. clustering coefficient, efficiency, etc.).*

Multivariate modeling and predictive analysis

*Specify independent variables, features extraction and dimension reduction, model, training and evaluation metrics.*

