## [Peer Review File · Nature Genetics]

Peer Review Information

Manuscript Title: Accurate detection of identity-by-descent segments in human ancient DNA

Corresponding author name(s): Dr Harald Ringbauer, Dr David Reich

Editorial Notes:

Unpublished Data: The originally submitted version of this manuscript and the figures in this TPR file used a larger set of unpublished data to show the ability of ancIBD to detect relatives up to the 6th degree. The authors have used only the published subset in the accepted version, which shows qualitatively similar results and hence does not affect the conclusions drawn.

Reviewer Comments & Decisions:

Decision Letter, initial version:

11th May 2023

Dear Harald,

Your Article, "ancIBD - Screening for identity by descent segments in human ancient DNA" has now been seen by 2 referees. You will see from their comments below that while they find your work of considerable interest, some important points are raised. We are very interested in the possibility of publishing your study in Nature Genetics, but would like to consider your response to these concerns in the form of a revised manuscript before we make a final decision on publication.

In summary, the two referees present extremely enthusiastic reports, suggesting there is not a major amount of additional work required before your study is ready for publication. Importantly, and pleasingly, they state that ancIBD will be of broad interest and utility. In our reading of each referee's specific requests, we did not think there was anything that seemed unduly onerous or requiring substantial further work; and we felt that each referee's suggestions on further illustrative analyses (#1) or to improve the usability of ancIBD (#2) are useful ones and would encourage you and your co-authors to fulfil them to the best of your capability.

To guide the scope of the revisions, the editors discuss the referee reports in detail within the team,

including with the chief editor, with a view to identifying key priorities that should be addressed in revision and sometimes overruling referee requests that are deemed beyond the scope of the current study. We hope that you will find the prioritized set of referee points to be useful when revising your study. Please do not hesitate to get in touch if you would like to discuss these issues further.

We therefore invite you to revise your manuscript taking into account all reviewer and editor comments. Please highlight all changes in the manuscript text file. At this stage we will need you to upload a copy of the manuscript in MS Word .docx or similar editable format.

*2) If you have not done so already please begin to revise your manuscript so that it conforms to our Article format instructions, available [here](http://www.nature.com/ng/authors/article_types/index.html). Refer also to any guidelines provided in this letter.

[redacted]

Nature Genetics is committed to improving transparency in authorship. As part of our efforts in this direction, we are now requesting that all authors identified as 'corresponding author' on published papers create and link their Open Researcher and Contributor Identifier (ORCID) with their account on

the Manuscript Tracking System (MTS), prior to acceptance. ORCID helps the scientific community achieve unambiguous attribution of all scholarly contributions. You can create and link your ORCID from the home page of the MTS by clicking on 'Modify my Springer Nature account'. For more information please visit www.springernature.com/orcid.

Sincerely,

Michael Fletcher, PhD
Senior Editor, Nature Genetics

ORCID: 0000-0003-1589-7087

Referee expertise:

Referee #1: human and statistical genetics.

Referee #2: ancient DNA; human genetics and evolution.

Reviewers' Comments:

Reviewer #1:

Remarks to the Author:

The paper discusses the significance of Identical by Descent (IBD) segments in identifying biological relatives. Existing methods for calling IBD segments in present-day genomes cannot be directly applied to ancient DNA due to low coverage and high genotyping error rates. The authors present a new method, called ancIBD, which utilizes a Hidden Markov Model to make inference from genotype imputation posteriors. Through simulations, the authors demonstrate that ancIBD reliably identifies IBD segments longer than 8 cM in aDNA data. The method is applied to analyze ancient Eurasian individuals, leading to the identification of distant relatives and evidence of long-distance migration. Additionally, ancIBD reveals insights into the spread of Steppe pastoralist ancestry into Europe, including a strong genetic connection between Corded Ware and Yamnaya individuals.

The aDNA community will find this work relevant. The ability of ancIBD to identify IBD segments in ancient genomes should allow researchers to explore genetic connections at various scales. The method's applications, demonstrated through simulated and real data, showcase its efficacy in revealing ancient familial relationships and shedding light on large-scale historical events.

Overall, this article demonstrates technical competence, and there are no significant flaws that cast doubt on the presented results. We do have some comments regarding method limitations, result interpretation, and the conducted simulations. In addition to these points, we have some suggestions to enhance the manuscript. However, it should be noted that the article is well-written, includes

informative visualizations, and is generally easy to comprehend. From what we can ascertain, relevant research has been appropriately cited.

A - Major comments

A1. In Supplementary Note S2, the authors provide a description of the methodology used to generate the validation dataset presented in Fig. 2. However, we have several concerns regarding the process employed to make the shuffled data resemble ancient data. Firstly, the explanation provided lacks clarity, and we would suggest rephrasing it or providing a schematic figure to illustrate the approach. Secondly, it is not evident how effective this approach is in producing imputed ancient-like data. Therefore, we would ask the authors to provide a detailed explanation along with a clear demonstration of how the imputed ancient-like data is obtained during the validation step.

A2. ancIBD treats adjacent or overlapping IBD segments as a single, longer IBD segment. Although you acknowledge that IBD sharing beyond a pair of haplotypes is rare in practice, it is actually common among close relatives such as siblings (~25% of the genome). In such cases, the number of IBD segments would be underestimated, while the length of the IBD segments would be overestimated. Related to this, distinguishing between IBD-0, IBD-1, and IBD-2 regions shared between pairs of individuals is at the basis of kinship inference (as highlighted by Popli et al., 2023). It seems to us that ancIBD is unable to perform such a distinction. To prevent the algorithm from being misapplied by the scientific community, it is essential to emphasize these limitations (introduction/discussion).

A3. While the authors did consider WGS data in their study, it would have been informative for the community to delve deeper into the advantages and disadvantages of each approach in terms of IBD mapping. Throughout the paper, several results indicate that WGS outperforms the 1240k data significantly when matched for coverage depth. Some more elaborated discussions on this would help in understanding these findings. Why does WGS perform better? Is that related to the total amount of available sequence and not the depth at considered positions? (i.e. 1x of capture is less sequence data than 1x of WGS). Why are correctly phased blocks so much longer? (Table S2)? Additionally, it would be useful to compare the performance of WGS with and without the 1240k subsetting, specifically assessing the impact of SNP density on IBD detection. In other words, examining the performance of WGS when utilizing all common variations would provide valuable insights.

B - Suggested analyses (optional)

B1. Fig. 3a: In order to perform a meaningful comparison between the IBD estimates derived from the 1240K dataset and those obtained using the simulated pairs, it would be beneficial to have color labels in Figure 3a, similar to the ones provided in Figure 3b. While we acknowledge that determining kinship for ancient samples may not be feasible beyond a certain degree of relatedness, it would still be valuable to include any available kinship information for the ancient samples whenever possible. Additionally, drawing the boundaries of the clusters identified in Figure 3b onto Figure 3a would enhance the ability to compare the results more effectively.

C - Minor comments

C1. When testing ancIBD, we had to make some modifications to make the vignette

"create_hdf5_from_vcf" work: i. we had to modify the prepare_h5.py file so that it writes the temporary .vcf filtered for 1240k to an uncompressed format (replace -Oz by -Ov) and ii. we had to change the variable "marker_path" (set map_path = f"./data/afs/v51.1_1240k.snp")

C2. In the emission probabilities part, when you use the allele frequencies in a population, do you use the overall allele frequencies in the 1000 Genomes panel, or population-specific allele frequencies (EUR, AME, AFR...)? What is the impact of using population-specific allele frequencies?

C3. The description of the imputation pipeline in Supplementary Note S3 lacks sufficient details. Specifically, the authors mention a study that introduces an imputation method for pseudo-haploid data. It is apparent from the mentioned manuscript that glimpse was used, which requires the generation of genotype likelihoods using a ploidy of 2 for the 1240K dataset. To improve this section, could you please explicitly state the generation of genotype likelihoods with a ploidy of 2 from the 1240K dataset and provide more details on the imputation steps?

Furthermore, it would strengthen the study if a fully imputed version of the 1240K dataset could be made available as a resource. Are there any plans to make this fully imputed dataset accessible to the community?

C4. We have a couple of comments regarding Fig. 2b. Firstly, there seems to be an issue with the y-axis label, which is currently stated as "number of FP Segments per Pair of ch3." However, considering that the scale ranges from 0 to approximately 0.15, it appears that this should be represented as a fraction instead. Could you please clarify if the y-axis label should indeed be a fraction?

Secondly, we would appreciate further clarification on the interpretation of the expected IBD curves with different effective population sizes. Specifically, is the scale on the y-axis the same for both the false positive rates and the expected IBD distributions? It would be helpful to explain the purpose behind contextualizing the false positive rate with the expected IBD. What is the objective in relating these two measures?

C5. The plots in figure 3a and 3b do not have the same scale on the y-axis.

C6. Readers would greatly benefit from additional details on the imputation pipeline employed in this study. The supplementary materials only offer a brief four-line description of the imputation process. It would be beneficial to include information such as the number of variants imputed, their minor allele frequency (whether they were all imputed or just a subset), the glimpse version used, and other pertinent details. As this step is crucial for ancIBD, elaborating on the procedure would aid readers' comprehension.

C7. A benchmarking of your approach on different levels of relationships would be welcome. For instance, by comparing the performance of your algorithm on avuncular pairs vs cousins etc... (as in Popli et al. 2023).

Reviewer #2:

Remarks to the Author:

Ringbauer et al. present a computational method called ancIBD to detect genetic relatives in ancient DNA data, through the identification of long shared IBD tracts. They also apply this new method on available human data, making several interesting findings that contribute to our understanding of human population history and demonstrate the power of the method. The identification of relatives has quickly become a major topic of interest in the human ancient DNA field, as it adds a whole other

level of information on top of traditional methods that quantify genetic similarity in more indirect ways. The work presented in this paper certainly represents the very cutting edge in this space, and it will undoubtedly be very influential in this research field.

The computational method appears to be very well-implemented, very robust validation results are provided, the paper is well-written, and I have no substantial concerns or complaints. I would strongly recommend this paper for publication, and I don't think much further work is needed on it. In what follows I make various comments and suggestions to the authors, but none of these would in my view constitute a reason to not publish the paper in Nature Genetics. Many of these relate to the software itself and how it's presented, and how users might interact with it, rather than the manuscript per se.

1) The method is focussed on human data, as even the title makes clear. In principle, the method could be applied to other species, and I would argue that the best methods in population genetics are made to be species-agnostic. Users can then choose arguments and parameters that suit their species of interest (such as the ones the authors identify as suitable for humans). If the authors here have made the decision to only cater to human data, I suppose that's their choice and that's fine. But it would be an even stronger contribution if the software is indeed species-agnostic and more widely usable.

An important aspect of this is: Is any human data hard-coded into the software? E.g. chromosome numbers/names/lengths, the genetic map, or other things. This was not entirely clear to me, and it would be useful if this could be clarified. Ideally these kind of things should be choices made by the user (the user might also want to e.g. use different genetic maps, reference genomes etc.).

A minor point relating to this: "each of the 22 autosomal chromosomes" – is there anything stopping the user from running ancIBD on female X chromosomes?

2) I think a clearer presentation of the workflow of ancIBD would be useful, and surely serve the many future users of the software. I find it a bit challenging to understand what's part of the software and what's upstream of it. For example, is the calculation of genotype likelihoods using BCFTools and the phasing using GLIMPSE considered part of ancIBD – that is, ancIBD runs wrappers around these? Or, are these upstream steps that the users need to run themselves? The latter would probably be preferable, as it would mean greater flexibility and allow users to make their own choices, for example if a better phasing method becomes available. Likewise, does a user have to use the 1000 Genomes reference panel, or can they make their own choice (there are bigger panels available at this point)? This was not really clear to me.

A smaller point relating to this: "In practice, we obtain p from the allele frequencies in the reference panel." This concerns calculations within the ancIBD software, but here ancIBD appears to access information from the phasing reference panel. Judging from the flowchart in Figure 1B, the 1000G reference panel only feeds information into GLIMPSE, not into ancIBD. It's thus not clear to me how the allele frequency information flows here, and whether a user needs to have the reference panel accessible on disk in order to run ancIBD, or if they in principle can use pre-phased files.

3) I wonder about the user-friendliness of the software. It's run as Python package, and judging from the online documentation it appears that users have to write short Python scripts to run it. While this is nothing out of the ordinary, I think it will reduce the usability of the software to biologists with less computational skills. In principle I think ancIBD could become the killer app for the whole field, but

currently it seems like running the software might just be slightly too involved for it to become that. I wonder if it would be feasible to provide a command-line wrapper version that would enable many common use cases directly on the command line, while leaving the full package for the more sophisticated users. This is merely a suggestion of course, but I think it could bring the software to a place where it can see really wide usage.

Various minor comments:

- "IBD longer than 8 centimorgan in aDNA data for SNP capture aDNA with" – sentence needs fixing
- "Applying genomic masks" – is this mask applied by default? Can the user supply their own mask file?
- "ancIBD generally takes as input imputed 1240k sites" – not clear what "generally" means here, does it mean "by default"? Is the input sites not up to the user?
- "Consortium et al., 2015" – check formatting of the 1000 Genomes citation

Author Rebuttal to Initial comments

Response to referees

We are delighted to have received thoughtful and supportive reviews for our manuscript. We have addressed the suggestions provided by the reviewers and made substantial revisions. These include additional illustrative experiments and a more comprehensive description of our method, *ancIBD*. We believe that these updates, highlighted in red text where possible, have enhanced the quality of our work, and hope that they render it suitable for publication in *Nature Genetics*.

In addition to the manuscript revisions, we have also made major improvements to the Python package that implements the method described in our paper (*ancIBD*) and have also expanded its public online documentation.

To provide a detailed account of our response to the reviewers' comments, please find the point-by-point responses attached below. We include the original reviewers' comments as well as our responses, which are highlighted in green.

Reviewer #1

Remarks to the Author:

The paper discusses the significance of Identical by Descent (IBD) segments in identifying biological relatives. Existing methods for calling IBD segments in present-day genomes

cannot be directly applied to ancient DNA due to low coverage and high genotyping error rates. The authors present a new method, called ancIBD, which utilizes a Hidden Markov Model to make inferences from genotype imputation posteriors. Through simulations, the authors demonstrate that ancIBD reliably identifies IBD segments longer than 8 cM in aDNA data. The method is applied to analyze ancient Eurasian individuals, leading to the identification of distant relatives and evidence of long-distance migration. Additionally, ancIBD reveals insights into the spread of Steppe pastoralist ancestry into Europe, including a strong genetic connection between Corded Ware and Yamnaya individuals.

The aDNA community will find this work relevant. The ability of ancIBD to identify IBD segments in ancient genomes should allow researchers to explore genetic connections at various scales. The method's applications, demonstrated through simulated and real data, showcase its efficacy in revealing ancient familial relationships and shedding light on large-scale historical events.

Overall, this article demonstrates technical competence, and there are no significant flaws that cast doubt on the presented results. We do have some comments regarding method limitations, result interpretation, and the conducted simulations. In addition to these points, we have some suggestions to enhance the manuscript. However, it should be noted that the article is well-written, includes informative visualizations, and is generally easy to comprehend. From what we can ascertain, relevant research has been appropriately cited.

A - Major comments

A1. In Supplementary Note S2, the authors provide a description of the methodology used to generate the validation dataset presented in Fig. 2. However, we have several concerns regarding the process employed to make the shuffled data resemble ancient data.

A1a) Firstly, the explanation provided lacks clarity, and we would suggest rephrasing it or providing a schematic figure to illustrate the approach.

We have improved the explanation of how we generated our simulated validation dataset. First, we expanded and restructured the text description in Section S2. Second, we have updated Fig. S1, as suggested by the reviewer, to now also include a graphic illustration of how we generated genotype probabilities matched to those that we empirically inferred when downsampling and imputing high-coverage ancient DNA:

A1b) Secondly, it is not evident how effective this approach is in producing imputed ancient-like data. Therefore, we would ask the authors to provide a detailed explanation along with a clear demonstration of how the imputed ancient-like data is obtained during the validation step.

In addition to an improved explanation of the simulated data set (see reply above) that should make clearer that we match aDNA imputation uncertainties by design, we now describe two experiments demonstrating that the simulated imputed ancient-like data accurately matches imputed empirical ancient data in key metrics (please see Supp Section S2.1).

A2. ancIBD treats adjacent or overlapping IBD segments as a single, longer IBD segment. Although you acknowledge that IBD sharing beyond a pair of haplotypes is rare in practice, it is actually common among close relatives such as siblings (~25% of the genome). In such cases, the number of IBD segments would be underestimated, while the length of the IBD segments would be overestimated. Related to this, distinguishing between IBD-0, IBD-1, and IBD-2 regions shared between pairs of individuals is at the basis of kinship inference (as highlighted by Popli et al., 2023). It seems to us that ancIBD is unable to perform such a distinction. To prevent the algorithm from being misapplied by the scientific community, it is essential to emphasize these limitations (introduction/discussion).

We now added a paragraph to the discussions, discussing that we focus on “IBD \geq 1” by choice, and why we believe that this is not a serious limitation for most IBD-based applications (e.g. that it is trivial to distinguish full-sibs from other biological relationships using IBD \geq 1 segments alone):

L502-L514 (main text):

Our algorithm infers the presence of at least one shared IBD segment between two diploid individuals, but in practice both pairs or even three or all four haplotypes can be shared. Here, we kept HMM's hidden state space simple to improve robustness and runtime. Importantly, we believe detecting the presence of one IBD segment alone suffices for most practical applications. Double IBD sharing occurs mostly in full siblings, who on average share half of their genome length in a single IBD and one additional quarter in a double IBD. In this case, the sum of IBD length alone distinguishes full siblings from parent-offspring pairs (who distinctively have their whole genome in IBD) and second-degree relatives (separate clusters in Fig.S21). Beyond full siblings, having overlapping IBD segments on different haplotype pairs only rarely occurs in practice [Chiang et al., 2016]. Only in special cases, such as distinguishing double first cousins from other second-degree relatives, identifying double IBD can be useful. Then, we recommend directly screening for identical imputed genotypes in IBD segments.

A3. While the authors did consider WGS data in their study, it would have been informative for the community to delve deeper into the advantages and disadvantages of each approach in terms of IBD mapping. Throughout the paper, several results indicate that WGS outperforms the 1240k data significantly when matched for coverage depth. Some more elaborated discussions on this would help in understanding these findings. Why does WGS perform better? Is that related to the total amount of available sequence and not the depth at considered positions? (i.e. 1x of capture is less sequence data than 1x of WGS). Why are correctly phased blocks so much longer? (Table S2)?

As the reviewers suggest, the improved performance of WGS is mainly due to imputation using variants not on the 1240k SNP set of which WGS data has vastly more. To demonstrate this, we now directly show imputation quality - revealing that WGS data can be imputed at 3x lower coverage on the SNP target as 1240k data while maintaining similar performance (Fig. S5). Similar imputation performance differences between WGS and 1240k data have also been noted in a recent study of GLIMPSE-imputed aDNA (Mota et al., 2023), where they report that “for the same individual samples, the imputation performances of 1x capture and shotgun-sequenced data with depth of coverage between 0.1x and 0.5x were

equivalent”.

To clarify this to the reader, we added a paragraph in the discussion:

L449-L459 (main text):

Our benchmarks have demonstrated that ancIBD robustly detects IBD longer than 8-cM, for WGS data down to 0.25x and 1240k data down to 1x average coverage depth on 1240k SNPs. That WGS data performs better than 1240k data at the same coverage depth on target SNPs is not surprising because WGS data covers the entire genome while 1240k capture data is depleted for off-target data. But imputation at 1240k sites uses all SNPs in the 1000 Genome dataset, thus providing more off-target data leads to substantially improved imputation quality. We found that WGS data can be imputed at roughly three times lower coverage equally well as 1240k data (see Fig.S5), consistent with findings from Sousa da Mota et al., 2013. This observation is relevant for choosing aDNA data generation strategies where IBD segment calling is of interest.

Additionally, it would be useful to compare the performance of WGS with and without the 1240k subsetting, specifically assessing the impact of SNP density on IBD detection. In other words, examining the performance of WGS when utilizing all common variations would provide valuable insights.

This is a valuable suggestion. We conducted this experiment and describe and summarize our findings in Supp Note S6 and Fig S8 :

L254-L273 (supplement):

We explored whether using all common variants in the 1000Genome variant sets can improve the performance of ancIBD. We filtered to 1000Genome SNPs with minor allele frequency (MAF) greater than 5% and adjusted the parameter “snp_cm” (minimum SNP density per cM within IBD segments) from the default 220 to 800 because there are about six times more 1000G SNPs with MAF>5% than 1240k SNPs. We tested this expanded SNP set both on WGS and on 1240K aDNA data. For WGS aDNA data using 1000G SNPs with MAF>5% resulted in slightly improved performance for shorter IBD segments (6-8cM) while for long segments (>12cM) the precision and recall remain almost identical (Fig.S8). For 1240k aDNA data using 1000G SNPs with MAF>5% gave mixed results (Fig.S8). In particular, for longer segments (>8cm), we observe a marked reduction in recall compared to when using 1240K SNPs.

As the benefits of utilizing all imputed common variants in 1000G SNPs are limited to WGS data and generally small, we recommend that ancIBD is run on data filtered to the 1240k

SNP set after imputation. This has the practical benefit of increasing the co-analyzability of 1240k and WGS data, by having a single standard pipeline that can be applied to both kinds of data and also mixes thereof. However, we note that \ancIBD can be run on any SNP set that is provided as input data, users can in principle choose and experiment with other SNP sets.

B - Suggested analyses (optional)

B1. Fig. 3a: In order to perform a meaningful comparison between the IBD estimates derived from the 1240K dataset and those obtained using the simulated pairs, it would be beneficial to have color labels in Figure 3a, similar to the ones provided in Figure 3b. While we acknowledge that determining kinship for ancient samples may not be feasible beyond a certain degree of relatedness, it would still be valuable to include any available kinship information for the ancient samples whenever possible.

Following the suggestion of the reviewer, we now color-code Fig. 3a using a conventional method to find relatives based on pairwise mismatch rates (PMR). We match the color coding of panel 3b. We added a new section to the Supplement to describe how we estimate PMR-based relatedness and identify up to including the third degree (Supp. Note S9).

Additionally, drawing the boundaries of the clusters identified in Figure 3b onto Figure 3a would enhance the ability to compare the results more effectively.

We have explored bringing the boundaries over (see figure below, the boundaries are drawn based on the mean and covariance of each simulated relative class). Unfortunately, we believe that this overloads this figure, in particular now that we also color-code Figure 3a by pairwise mismatch rates. However, we have now adjusted the plot to share the same Y-axis to make the two panels visually directly comparable (see also reply C5 below).

C - Minor comments

C1. When testing ancIBD, we had to make some modifications to make the vignette “create_hdf5_from_vcf” work: i. we had to modify the prepare_h5.py file so that it writes the temporary .vcf filtered for 1240k to an uncompressed format (replace -Oz by -Ov)

Thank you for this bug report and a suggested fix. Indeed, the problem was an incompatibility occurring on some systems when running bcftools with compressed output (-Oz) when the actual temporary VCF path file ended in “.vcf” (and not in “.vcf.gz”). Our updated code now checks for the ending and runs compression only if the output file has “.gz” suffix otherwise, we produce an uncompressed intermediate file as suggested (using the flag -Ov). This fix is now included in the latest ancIBD release (version 0.5).

and ii. we had to change the variable “marker_path” (set map_path = f"./data/afs/v51.1_1240k.snp")

Thank you for spotting the incorrect path in our vignette. We have now moved the .snp file into a ./map folder to avoid any confusion and changed the path variables in our online tutorial accordingly.

C2. In the emission probabilities part, when you use the allele frequencies in a population, do you use the overall allele frequencies in the 1000 Genomes panel, or population-specific allele frequencies (EUR, AME, AFR...)? What is the impact of using population-specific allele frequencies?

That is an insightful comment, indeed ancIBD uses allele frequencies in the emission model and we have not described our choice in the original manuscript. We now describe the choice of allele frequencies (using generally the 1000G allele frequencies that are coded into the output .vcf of GLIMPSE) in a new Supplementary section (Supp. Note S4, L140-L160). Moreover, we show there that ancIBD is robust to choice of allele frequencies. A new experiment using either the imputed sample allele frequency of all Eurasian ancient individuals or the 1000 Genome allele frequencies shows highly correlated results for IBD calls of *ancIBD* (Fig.S6).

C3. The description of the imputation pipeline in **Supplementary Note S3** lacks sufficient details. Specifically, the authors mention a study that introduces an imputation method for pseudo-haploid data. It is apparent from the mentioned manuscript that GLIMPSE was used, which requires the generation of genotype likelihoods using a ploidy of 2 for the 1240K dataset. To improve this section, could you please explicitly state the generation of genotype likelihoods with a ploidy of 2 from the 1240K dataset and provide more details on the imputation steps?

We have substantially improved Supp Note S3. We include now the bash commands used to generate the genotype likelihoods (of ploidy 2) and we expanded the description of how we imputed this data with GLIMPSE. We highlight now that for steps after the genotype likelihood calling, we exactly followed the official tutorial of GLIMPSE, and we refer readers to this GLIMPSE tutorial.

Furthermore, it would strengthen the study if a fully imputed version of the 1240K dataset could be made available as a resource. Are there any plans to make this fully imputed dataset accessible to the community?

We fully agree that publishing the imputed data will be a great resource, beyond the application of calling IBD. A subset of the current manuscript's authors is working on a separate publication that focuses on the imputed dataset and a more refined analysis of the imputed data, beyond the focus on the performance of IBD calling. That publication will also showcase downstream applications and visualizations (such as tracking allele frequencies through time and space). It will be the avenue to publish the imputed data set - and will give the proper credit to the co-authors that curated and created the imputed data set. We note that all our inferences on published aDNA data with ancIBD are fully replicable from the original bam files.

C4. We have a couple of comments regarding Fig. 2b. Firstly, there seems to be an issue with the y-axis label, which is currently stated as "number of FP Segments per Pair of ch3." However, considering that the scale ranges from 0 to approximately 0.15, it appears that this should be represented as a fraction instead. Could you please clarify if the y-axis label should indeed be a fraction?

Thanks for pointing out that this figure caption lacked clarity. We have now changed the y-axis label and added an explanation in the figure caption to clarify that the y-axis is the average number of FP segments on chromosome 3 per pair.

Secondly, we would appreciate further clarification on the interpretation of the expected IBD curves with different effective population sizes. Specifically, is the scale on the y-axis the same for both the false positive rates and the expected IBD distributions? It would be helpful to explain the purpose behind contextualizing the false positive rate with the expected IBD. What is the objective in relating these two measures?

The FP and the expected IBD curves are plotted using the same scale. To clarify the purpose of this comparison, we have added the following to the figure caption of Fig.2:

Caption for Fig 2b:

To contextualize these false positive rates, we also depict expected IBD sharing assuming various constant population sizes (dotted lines, calculated as described in Fernandes et al. [2021]). If the false positive rate is on a similar order of magnitude or larger than expected for a population of that effective size, individual IBD calls of that length for that coverage and demographic scenario are likely to be false positives.

C5. The plots in Figure 3a and 3b do not have the same scale on the y-axis.

We have adjusted the y-axis limits of Fig. 3a and 3b, they now use the same scale. We note that we have added the same post-processing to simulated IBD segments as by default in ancIBD, in particular filtering IBD segments with low SNP density (that are prone to false positives in practice) - we describe this in Supp. Note S8. This post-processing makes the two panels now very directly comparable.

C6. Readers would greatly benefit from additional details on the imputation pipeline employed in this study. The supplementary materials only offer a brief four-line description of the imputation process. It would be beneficial to include information such as the number of variants imputed, their minor allele frequency (whether they were all imputed or just a subset), the glimpse version used, and other pertinent details. As this step is crucial for ancIBD, elaborating on the procedure would aid readers' comprehension.

We have substantially expanded and clarified the description of imputation, see the response to point C3 above.

C7. A benchmarking of your approach on different levels of relationships would be welcome. For instance, by comparing the performance of your algorithm on avuncular pairs vs cousins etc... (as in Popli et al. 2023).

We believe that benchmarking relationship tools in ancient DNA is a complex task, as the performance will depend on the level of background relatedness and genetic diversity within a population and also between the putative relatives, and moreover on data quality (aDNA damage, etc.). We, therefore, believe that this is best explored by dedicated future work.

However, we have expanded our discussion, where we highlight the evident advantages of both methods, in particular, that ancIBD is currently the only tool for ancient DNA that can detect IBD segments that identify relatives more distant than third-degree:

L461-L469:

First, ancIBD reveals biological relatives up to the sixth degree as such pairs distinctively share multiple long IBD segments [Caballero et al., 2019]. Allele-sharing-based methods commonly used in aDNA studies [Lipatov et al., 2015, Monroy Kuhn et al., 2018] are generally limited to detecting relatives only up to the third degree because they average over the genome and do not identify signals due to only a few shared IBD segments that make up only a small part of the genome. However, they can be applied to substantially lower coverage than ancIBD. Similarly, KIN [Popli et al., 2023] can be applied to lower coverage than ancIBD, but is also limited to detecting relatives up to the third degree.

Reviewer #2:

Remarks to the Author:

Ringbauer et al. present a computational method called *ancIBD* to detect genetic relatives in ancient DNA data, through the identification of long shared IBD tracts. They also apply this new method on available human data, making several interesting findings that contribute to our understanding of human population history and demonstrate the power of the method. The identification of relatives has quickly become a major topic of interest in the human ancient DNA field, as it adds a whole other level of information on top of traditional methods that quantify genetic similarity in more indirect ways. The work presented in this paper certainly represents the very cutting edge in this space, and it will undoubtedly be very influential in this research field.

The computational method appears to be very well-implemented, very robust validation results are provided, the paper is well-written, and I have no substantial concerns or complaints. I would strongly recommend this paper for publication, and I don't think much further work is needed on it. In what follows I make various comments and suggestions to the authors, but none of these would in my view constitute a reason to not publish the paper in Nature Genetics. Many of these relate to the software itself and how it's presented, and how users might interact with it, rather than the manuscript per se.

1) The method is focused on human data, as even the title makes clear. In principle, the method could be applied to other species, and I would argue that the best methods in population genetics are made to be species-agnostic. Users can then choose arguments and parameters that suit their species of interest (such as the ones the authors identify as suitable for humans). If the authors here have made the decision to only cater to human data, I suppose that's their choice and that's fine. But it would be an even stronger contribution if the software is indeed species-agnostic and more widely usable.

An important aspect of this is: Is any human data hard-coded into the software? E.g. chromosome numbers/names/lengths, the genetic map, or other things. This was not entirely clear to me, and it would be useful if this could be clarified. Ideally, these kind of things should be choices made by the user (the user might also want to e.g. use different genetic maps, reference genomes, etc.).

We did not hard-code any human-specific data into this software. Therefore, *ancIBD* can in principle be applied to any species. We note that we optimized the default parameters for human ancient DNA data, but all parameters (listed in Table.1) can be changed by the user.

We have to stress that ancIBD depends on imputation with a reference panel of phased haplotypes. Currently, imputation for other species is not a routine task as in human genetics, and in many cases even infeasible due to the lack of any suitable haplotype reference panel. We hope that this will change in the near future, as larger genomic resources become widely available for many species.

To highlight that ancIBD is species-agnostic, we want to briefly showcase work in progress on Rhesus Macaques. Together with collaborators, we have imputed low-coverage genomes against a reference panel of size ~750 Macaques and then applied ancIBD to call IBD. We have found excellent IBD calling performance even without parameter tuning (see the figure below). This application highlights the general usability of ancIBD for non-human species, given that the necessary genomic resources are available (i.e., a reference genome and a suitable reference panel for imputation).

Figure: ancIBD applied to a pair of Rhesus Macaque genomes. We called IBD in a pair of high-coverage genomes using the method IBIS (dark red IBD segments). We then downsampled those to ca. 4x average coverage. After imputation with a Macaque reference

panel, we applied ancIBD (orange IBD segments), obtaining excellent agreement with the ground truth data. Note that Macaques only have 20 autosomal chromosomes.

A minor point relating to this: “each of the 22 autosomal chromosomes” – is there anything stopping the user from running ancIBD on female X chromosomes?

Calling IBD between X chromosomes is generally an interesting extension of ancIBD. We have added a paragraph about this to the discussion:

L515-L523 (main text):

One promising extension is calling IBD segments on X chromosomes. Genetic males have only one copy of it, while females have two, which causes sex-specific inheritance and recombination patterns (e.g. males must have inherited their X chromosomes from their mothers). Therefore, IBD sharing on the X chromosome can provide information about sex-specific relatedness and demography [see e.g. Buffalo et al., 2016]. Our work here focused on the autosomes that make up the majority of the human genome; however, one can in principle apply ancIBD to imputed female X chromosomes. To call IBD on the X in pairs involving males, one could adapt the state space of ancIBD in a technically straightforward way.

2) I think a clearer presentation of the workflow of ancIBD would be useful, and surely serve the many future users of the software. I find it a bit challenging to understand what’s part of the software and what’s upstream of it. For example, is the calculation of genotype likelihoods using BCFTools and the phasing using GLIMPSE considered part of ancIBD – that is, ancIBD runs wrappers around these? Or, are these upstream steps that the users needs to run themselves? The latter would probably be preferable, as it would mean greater flexibility and allow users to make their own choices, for example, if a better phasing method becomes available. Likewise, does a user have to use the 1000 Genomes reference panel, or can they make their own choice (there are bigger panels available at this point)? This was not really clear to me.

We have updated Figure 1 and its legend to clarify that upstream imputation is not part of *ancIBD*. As the reviewer highlights, this design has the advantage that the user is free to use genotype likelihood methods, reference panels, and imputation software of their choice - and also benefit from future updates of those methods. We added to the discussion that future improvements in imputation can be easily integrated:

L499-L501:

We note that ancIBD takes imputed data as input, thus future improvements of imputation software or reference panels can be easily integrated by updating the pre-processing.

A smaller point relating to this: “In practice, we obtain p from the allele frequencies in the reference panel.” This concerns calculations within the ancIBD software, but here ancIBD appears to access information from the phasing reference panel. Judging from the flowchart in Figure 1B, the 1000G reference panel only feeds information into GLIMPSE, not into ancIBD. It’s thus not clear to me how the allele frequency information flows here, and whether a user needs to have the reference panel accessible on disk in order to run ancIBD, or if they in principle can use pre-phased files.

This is an important point. We now describe the choice of allele frequencies in a new Supplementary section (see reply to Reviewer #1, point 2c). By default, we use allele frequencies of the reference panel, which are saved into its output by GLIMPSE (.vcf field INFO/RAF). Therefore, using its output files (as depicted in Fig. 1b) is fully sufficient, users do not need to have the reference panel accessible on disk.

3) I wonder about the user-friendliness of the software. It’s run as Python package, and judging from the online documentation it appears that users have to write short Python scripts to run it. While this is nothing out of the ordinary, I think it will reduce the usability of the software to biologists with less computational skills. In principle I think ancIBD could become the killer app for the whole field, but currently it seems like running the software might just be slightly too involved for it to become that. I wonder if it would be feasible to provide a command-line wrapper version that would enable many common use cases directly on the command line, while leaving the full package for the more sophisticated

users. This is merely a suggestion of course, but I think it could bring the software to a place where it can see really wide usage.

Thank you for the suggestion of how to reach a wider audience with our tool. We have added shell command wrappers for *ancIBD* that are added by default when installing the package. Those are now described in the official documentation on *readthedocs* (<https://ancibd.readthedocs.io/en>) and are available as of ancIBD version 0.5.

Various minor comments:

- “IBD longer than 8 centimorgan in aDNA data for SNP capture aDNA with” – sentence needs fixing

Thank you for spotting this typo, we have fixed it.

- “Applying genomic masks” – is this mask applied by default? Can the user supply their own mask file?

Users can specify any mask files they have as optional input to *ancIBD*. Our masks are not applied by default. We have clarified these points in the FAQ section of our *readthedocs* page and have clarified this in writing:

L216-L217:

The human-specific masking is optional, the SNP 217 density filter is applied by default by *ancIBD*

- “*ancIBD* generally takes as input imputed 1240k sites” – not clear what “generally” means here, does it mean “by default”? Is the input sites not up to the user?

We have updated the relevant text section to clarify that we mean that we optimized the default parameters of *ancIBD* for using 1240k sites:

L219-L213:

In the following, we describe how we chose the default parameters of *ancIBD*. In principle, users can specify any SNP set as input, but our goal was to obtain default parameters that are optimized for imputed genotype likelihoods at the 1240k SNP set, as the majority of published human ancient DNA data consists of in-solution DNA capture experiments enriching for this SNP set.

- “Consortium et al., 2015” – check the formatting of the 1000 Genomes citation

Thank you for spotting that incorrect citation. We updated this citation to: “1000 Genomes Project Consortium, 2015”.

Decision Letter, first revision:

Our ref: NG-A62218R

11th Sep 2023

Dear Harald,

Thank you for submitting your revised manuscript "ancIBD - Screening for identity by descent segments in human ancient DNA" (NG-A62218R). It has now been seen by the original referees and their comments are below. The reviewers find that the paper has improved in revision, and therefore we'll be happy in principle to publish it in Nature Genetics, pending minor revisions to satisfy the referees' final requests and to comply with our editorial and formatting guidelines.

Thank you again for your interest in Nature Genetics. Please do not hesitate to contact me if you have any questions.

Sincerely,

Michael Fletcher, PhD
Senior Editor, Nature Genetics

ORCID: 0000-0003-1589-7087

Reviewer #1 (Remarks to the Author):

I am pleased to acknowledge that your responses effectively addressed all the points I had brought up. It is evident that you and your co-authors have taken my comments seriously.

Your clarity in explaining the revisions made it easy for me to follow the changes and see how they enhance the paper's overall quality.

I believe that the paper is on its way to making a valuable contribution to the field. I look forward to seeing the final version.

Reviewer #2 (Remarks to the Author):

I think the authors have satisfactorily responded to the comments from the reviewers, and have made changes that improve the clarity of the manuscript and probably also the ease with which users can run the software. I have no further comments and would recommend that the paper is published.

Final Decision Letter:

20th Oct 2023

Dear Harald,

I am delighted to say that your manuscript "Accurate detection of identity-by-descent segments in human ancient DNA" has been accepted for publication in an upcoming issue of Nature Genetics.

Your paper will be published online after we receive your corrections and will appear in print in the next available issue. You can find out your date of online publication by contacting the Nature Press Office (press@nature.com) after sending your e-proof corrections. Now is the time to inform your Public Relations or Press Office about your paper, as they might be interested in promoting its publication. This will allow them time to prepare an accurate and satisfactory press release. Include your manuscript tracking number (NG-A62218R1) and the name of the journal, which they will need when they contact our Press Office.

Please note that *Nature Genetics* is a Transformative Journal (TJ). Authors may publish their research with us through the traditional subscription access route or make their paper immediately open access through payment of an article-processing charge (APC). Authors will not be required to make a final decision about access to their article until it has been accepted. [Find out more about Transformative Journals](https://www.springernature.com/gp/open-research/transformative-journals)

Authors may need to take specific actions to achieve > **compliance with funder and institutional open access mandates**. If your research is supported by a funder that requires immediate open access (e.g. according to [Plan S principles](https://www.springernature.com/gp/open-research/plan-s-compliance)) then you should select the gold OA route, and we will direct you to the compliant route where possible. For authors selecting the subscription publication route, the journal's standard licensing terms will need to be accepted, including <https://www.nature.com/nature-portfolio/editorial-policies/self-archiving-and-license-to-publish>. Those licensing terms will supersede any other terms that the author or any third party may assert apply to any version of the manuscript.

An online order form for reprints of your paper is available at <https://www.nature.com/reprints/author-reprints.html>> <https://www.nature.com/reprints/author-reprints.html>. Please let your coauthors and your institutions' public affairs office know that they are also welcome to order reprints by this method.

If you have not already done so, we invite you to upload the step-by-step protocols used in this manuscript to the Protocols Exchange, part of our on-line web resource, natureprotocols.com. If you complete the upload by the time you receive your manuscript proofs, we can insert links in your article that lead directly to the protocol details. Your protocol will be made freely available upon publication of your paper. By participating in natureprotocols.com, you are enabling researchers to more readily reproduce or adapt the methodology you use. [Natureprotocols.com](http://natureprotocols.com) is fully searchable, providing your protocols and paper with increased utility and visibility. Please submit your protocol to <https://protocolexchange.researchsquare.com/>. After entering your [nature.com](http://www.nature.com) username and password you will need to enter your manuscript number (NG-A62218R1). Further information can be found at <https://www.nature.com/nature-portfolio/editorial-policies/reporting-standards#protocols>

Sincerely,

Michael Fletcher, PhD
Senior Editor, Nature Genetics

ORCID: 0000-0003-1589-7087